# Adaptation of *Bacillus thuringiensis* to Plant Colonization Affects Differentiation and Toxicity

Yicen Lin,[a] Monica Alstrup,[a] Janet Ka Yan Pang,[a] Gergely Maróti,[b] Mériem Er-Rafik,[c] Nicolas Tourasse,[d] Ole Andreas Økstad,[e,f] Ákos T. Kovács[a]

aBacterial Interactions and Evolution Group, DTU Bioengineering, Technical University of Denmark, Lyngby, Denmark
bInstitute of Plant Biology, Biological Research Centre of the Hungarian Academy of Sciences, Szeged, Hungary
cNational Centre for Nano Fabrication and Characterization, Technical University of Denmark, Lyngby, Denmark
dUniversité Bordeaux, CNRS, INSERM, ARNA, UMR 5320, U1212, Bordeaux, France
eCentre for Integrative Microbial Evolution, University of Oslo, Oslo, Norway
fSection for Pharmacology and Pharmaceutical Biosciences, Department of Pharmacy, University of Oslo, Oslo, Norway

**ABSTRACT** The *Bacillus cereus* group (*Bacillus cereus sensu lato*) has a diverse ecology, including various species that are vertebrate or invertebrate pathogens. Few isolates from the *B. cereus* group have however been demonstrated to benefit plant growth. Therefore, it is crucial to explore how bacterial development and pathogenesis evolve during plant colonization. Herein, we investigated *Bacillus thuringiensis* (Cry⁻) adaptation to the colonization of *Arabidopsis thaliana* roots and monitored changes in cellular differentiation in experimentally evolved isolates. Isolates from two populations displayed improved iterative ecesis on roots and increased virulence against insect larvae. Molecular dissection and recreation of a causative mutation revealed the importance of a nonsense mutation in the *rho* transcription terminator gene. Transcriptome analysis revealed how Rho impacts various *B. thuringiensis* genes involved in carbohydrate metabolism and virulence. Our work suggests that evolved multicellular aggregates have a fitness advantage over single cells when colonizing plants, creating a trade-off between swimming and multicellularity in evolved lineages, in addition to unrelated alterations in pathogenicity.

**IMPORTANCE** Biologicals-based plant protection relies on the use of safe microbial strains. During application of biologicals to the rhizosphere, microbes adapt to the niche, including genetic mutations shaping the physiology of the cells. Here, the experimental evolution of *Bacillus thuringiensis* lacking the insecticide crystal toxins was examined on the plant root to reveal how adaptation shapes the differentiation of this bacterium. Interestingly, evolution of certain lineages led to increased hemolysis and insect larva pathogenesis in *B. thuringiensis* driven by transcriptional rewiring. Further, our detailed study reveals how inactivation of the transcription termination protein Rho promotes aggregation on the plant root in addition to altered differentiation and pathogenesis in *B. thuringiensis*.

**KEYWORDS** *Arabidopsis thaliana*, *Bacillus thuringiensis*, experimental evolution, pathogenesis, plant-microbe interaction

Address correspondence to Ákos T. Kovács, atkovacs@dtu.dk.

**B**acillus cereus*, *Bacillus thuringiensis*, and *Bacillus anthracis* are the three most common species of the *Bacillus cereus sensu lato* group, which are Gram-positive and spore-forming organisms (1–3). Since they share highly similar 16S rRNA gene sequences, these species are distinguished based on the basic physiological and clinical phenotypic properties in addition to their plasmid contents (1, 2, 4). For instance, the presence of insecticidal crystal proteins is the common distinguishable feature of *B. thuringiensis* (5). Once these plasmids encoding crystal proteins are lost, *B. thuringiensis*

can no longer be distinguished from *B. cereus*. Over the last half century, *B. cereus* has been considered a common causative agent of food contamination responsible for diarrhea or emetic poisoning symptoms. The diarrheal syndrome is primarily caused by several enterotoxin proteins such as nonhemolytic enterotoxin (NHE), hemolysin BL (HBL), and cytotoxin K (CytK), encoded by chromosomal genes that are under the control of the global transcription regulator, PlcR (6–8). Cereulide peptide toxins are reported to be the major virulence factor responsible for emetic syndrome triggered by certain isolates (9, 10). Not surprisingly, these chromosomally encoded enterotoxins are shared by *B. thuringiensis*, and cases of human infections have been also reported from *B. thuringiensis* evasion (11, 12). *B. anthracis* cells produce capsules and harbor two major plasmids pXO1 and pXO2 encoding toxin proteins causing anthrax-like diseases which are considered one of the severest human infections (13). A nonsense mutation in the virulence regulator PlcR of *B. anthracis* prevents the expression of virulence genes that are otherwise transcribed in the genomes of other *B. cereus* group species (14). Previous studies have proposed *B. cereus* group bacteria to be regarded as a unique species consisting of exceedingly diverse strains differentiated by plasmid presence and expression of certain regulatory pathways (4, 15).

Apart from the focus on the clinical issue, massive research is related to the socioeconomic importance of *B. cereus* group species. *B. thuringiensis* has been successfully applied as a commercialized biopesticide for decades due to its entomopathogenic properties (16). As one of the most well-known product, it comprises approximately three quarters of the global bioinsecticide market (17). Mixtures of *B. thuringiensis* crystal proteins and spores can effectively control varieties of insects such as larvae of caterpillars, beetles, and nematodes (18). Several previous studies support the potent and specialized invertebrate pathogenicity of *B. thuringiensis* (16, 19–21). While the animal-associated environment of *B. thuringiensis* is well studied, other niches such as plant roots and soil are relatively less explored. Nevertheless, *B. thuringiensis* has also been described as plant growth-promoting bacteria (PGPB), suggesting that *B. thuringiensis* has the intrinsic ability to colonize plants, and the level is phylogeny dependent (22–24). In addition, various plant biocontrol products contain *Bacillus* spp., including *B. cereus* group organisms (25). Monnerat and colleagues demonstrated that when *B. thuringiensis* is inoculated into the soil, it could translocate throughout the plants and afterwards colonize the phylloplane stably (26). Recent evidence suggested that *B. thuringiensis* occurs in the plant rhizosphere that might play an important niche for the bacterium to access its animal hosts (20, 21). Similarly, *B. cereus* has also been isolated from the rhizosphere (27, 28). Plasmid-cured *B. anthracis* has been reported to germinate in the rhizosphere of plant seedlings and persist as vegetative cells (29). Thus, *B. thuringiensis* and other members of the *B. cereus* species have a complex ecological lifestyle and can be isolated as resistant spores or metabolically active vegetative cells from soil, rhizosphere, plant tissues, and living or dead insects.

However, the ecology and host range of *B. thuringiensis* are still not fully understood, especially due to its close taxonomic relatedness to other members of the *B. cereus* group that have public health concerns. Detailed understanding of its ecology will facilitate the application of *B. thuringiensis* in plant biologicals (23). Selection experiments can help us to reveal the mechanisms behind the different ecological traits of *B. thuringiensis* and whether or not this flexible microbe can specialize in certain environments. Besides traditional molecular genetics, experimental evolution (EE) is an effective approach for exploring the evolutionary dynamics of microbes in different environmental niches (30). From simple growth conditions such as shake flasks (31) and chemostat cultures (32) to complex environmental niches, including eukaryote hosts (33–35), EE has been successfully applied to microbes, revealing unique insights connecting microbial phenotypes and genotypes. Despite the prevalence of biofilms in nature, most EE has been performed using planktonic bacteria; hence, we lack a detailed understanding of adaptation within biofilm populations (36). A fascinating "bead transfer model" was developed to unravel the genetic diversification of the opportunistic

pathogen *Burkholderia cenocepacia* (37, 38). After long-term EE, *B. cenocepacia* diversified into different morphotypes with enhanced community productivity generated by niche complementarity effects (39). Similarly, diverse colony morphotypes arose during EE of highly wrinkled floating biofilms of *Bacillus subtilis*, called pellicles (40–43).

Compared with well-studied evolved biofilms of *B. subtilis*, knowledge on the *B. cereus* group remains limited. EE of *B. cereus* group strains has been performed within diverse hosts from nematodes to vertebrates to dissect host-microbe interactions and the evolution of virulence pathways (34, 44). Such approach demonstrated reciprocal coevolution between *B. thuringiensis* and its nematode host *Caenorhabditis elegans* (34), in contrast to one-sided adaptation that favored mutational landscape changes in certain toxin genes. Laboratory evolution of *B. cereus* group biofilms remains limited, despite the relevance of biofilms in various environmental niches, where the evolutionary ecology of virulence can also be monitored (45).

Herein, we devised an EE setup associating *B. thuringiensis* (Cry⁻) with the model plant *Arabidopsis thaliana* to scrutinize the parallel evolution of this bacterium. The Cry⁻ derivative of *B. thuringiensis* is a model strain resembling the properties of *B. cereus* isolates. After 40 cycles of laboratory evolution on the plant root, bacterial lineages displayed enhanced root colonization ability compared with the ancestral strain. Intriguingly, single isolates from two of the evolved lineages tended to recolonize a new root more efficiently compared with the other lineages, in addition to exhibiting altered bacterial differentiation and pathogenicity. Investigation of a key mutation in the gene encoding the Rho transcription termination factor in these lineages demonstrated how transcriptional rewiring alters cell fate decisions in *B. thuringiensis*.

## RESULTS

**Experimental evolution on plants selects for improved root colonizers.** EE of *B. thuringiensis* 407 Cry⁻ (Bt407) was initiated using six parallel lineages colonizing the roots of 7-day-old *A. thaliana* seedlings under hydroponic conditions generally applied for *Bacilli* (46–50). Initially, inoculated planktonic bacterial cells developed biofilms on plant roots. Subsequently, these biofilms were reestablished on new roots after transplanting seedlings to a new culture after 2 days (Fig. 1a). Plant-associated biofilms formed on roots were quantitatively monitored by assessing the productivity (bacterial CFU/root length) after every fifth transfer. Notably, one of the six lineages (lineage B) yielded CFU values below the detection limit after five transfers and was therefore removed. We speculate that the strong population bottleneck (i.e., only a few cells reestablishing a new biofilm) led to the disappearance of this lineage, since only a few hundred bacterial cells were attached to roots during the initial stages (Fig. 1b). Although these ecosystems were independent, biofilm productivity increased gradually in the other five populations throughout the whole EE period. The increased biomass provides evidence for successful adaptation during EE, as reported in previous studies (37, 43).

Unsurprisingly, evolved strains formed more extensive root biofilms than the Bt407 ancestor, several times thicker in places (see Fig. S1 in the supplemental material). To test the adapted traits, three colonies were isolated from each lineage after the 40th transfer and examined for biofilm establishment on roots. The majority of the evolved isolates exhibited significant root colonization enhancement compared to the ancestor strain (Fig. 1c).

Next, colonized plants with each of the three isolates from the evolved lineages were transferred to fresh medium hosting new seedlings, and CFU values were quantified. This capacity, termed as root recolonization, was strikingly increased in two of the evolved lineages (lineages E and F; Fig. 1d). When a new seedling was provided, the biofilm cells of lineages E and F tended to detach from the older seedlings and colonize the new ones more efficiently.

**Shifts in multicellular behavior accompany adaptation to root colonization.** Colonization of the rhizosphere by *Bacilli* depends on various multicellular traits (46, 51, 52). We specifically examined the evolved isolates from lineages E and F and

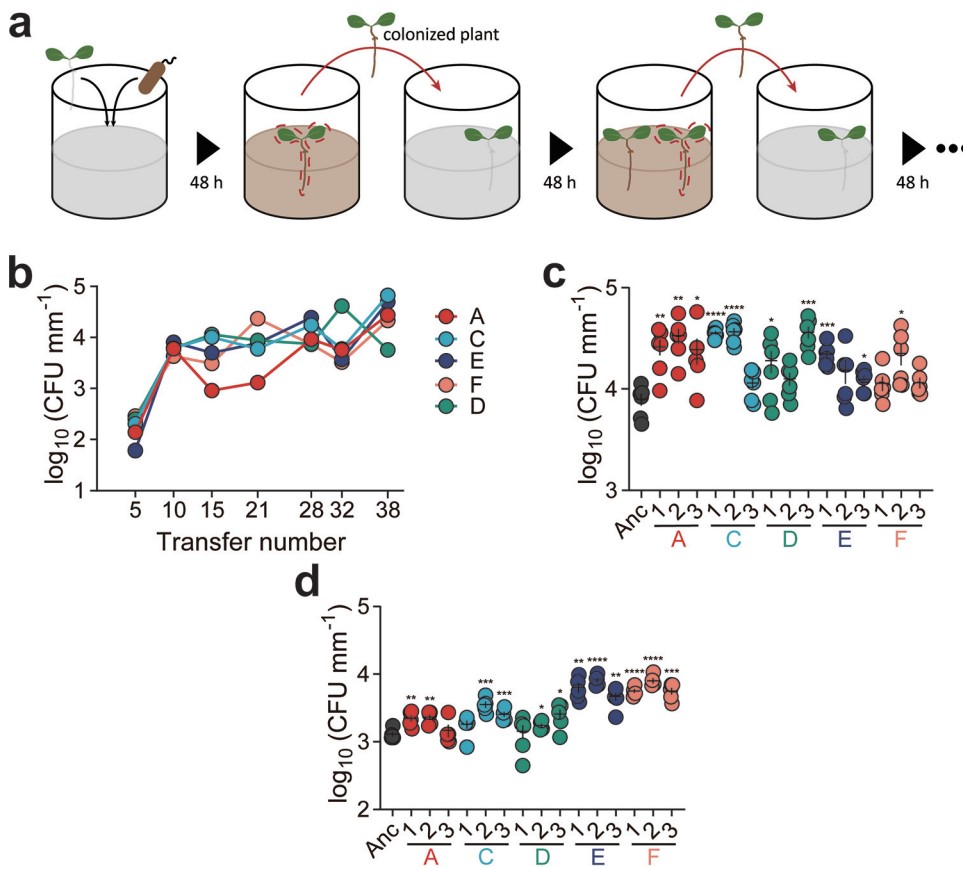

**FIG 1** Experimental evolution setup and productivity assessment. (a) Experimental evolution model of root-associated biofilms. Parallel lineages of the Bt407 (Cry⁻) strain were initiated on a 1-week-old *A. thaliana* seedling to form biofilm in MSNg medium in 48-well microliter plates agitated at 90 rpm. Colonized plants were subsequently transferred to a well containing a sterile seedling, and the steps were repeated for 40 transfers. (b) Productivity (CFU mm⁻¹) of plant-colonized Bt407 (Cry⁻) lineages are shown at roughly every fifth transfer. (c) Plant-colonized biofilm productivity (CFU mm⁻¹) of evolved isolates (*n* = 6 biologically independent plantlet samples of similar length). The central values (horizontal lines) represent the means, and the error bars represent standard errors of the means. Asterisks indicate significant differences between the values for each group and the ancestor (Anc) (*, *P* < 0.05; **, *P* < 0.01; ***, *P* < 0.001; ****, *P* < 0.0001; two-tailed *t* test with Welch's corrections). (d) Recolonized biofilm productivity (CFU mm⁻¹) of the evolved isolates (*n* = 6 biologically independent plantlet samples of similar length). Statistical significance assessment was conducted as described above for panel C.

compared the differentiation properties to the ancestor and the isolates from other evolved lineages (lineages A, C, and D). Primarily, colonization of a new niche depends on bacterial motility (53). *B. thuringiensis* displays two types of flagellum-driven motility: single cells swimming and swarming by surfactin-aided rafts of cells. Surprisingly, isolates evolved from the E and F lineages were greatly reduced in swimming motility, while other lineages exhibited motility comparable with that of the ancestor (Fig. 2a, Fig. S2A, and Fig. S3A). Swimming motility is required for air-liquid interface biofilm development of *B. cereus* in 24-h cultures (54), whereas under flow conditions, nonmotile cells form dense and thick biofilms. Notably, EE was performed using mildly shaken cultures, possibly facilitating random contact between bacterial debris and seedlings. Thus, swimming might not provide a benefit for bacteria to recolonize plants in this setup, and there might be a fundamental trade-off between the free-swimming state and community-based surface spreading. In contrast, when surface spreading was tested using increased agar concentration (i.e., 0.7% agar) to determine swarming, the isolates from lineages E and F displayed enhanced swarming compared to the ancestor (Fig. 2b, Fig. S2B, and Fig. S3B). Studies have shown that defects in swarming ability can cause poor root colonization in *Bacilli* (55, 56). Swarming cells suppress cell division and cell

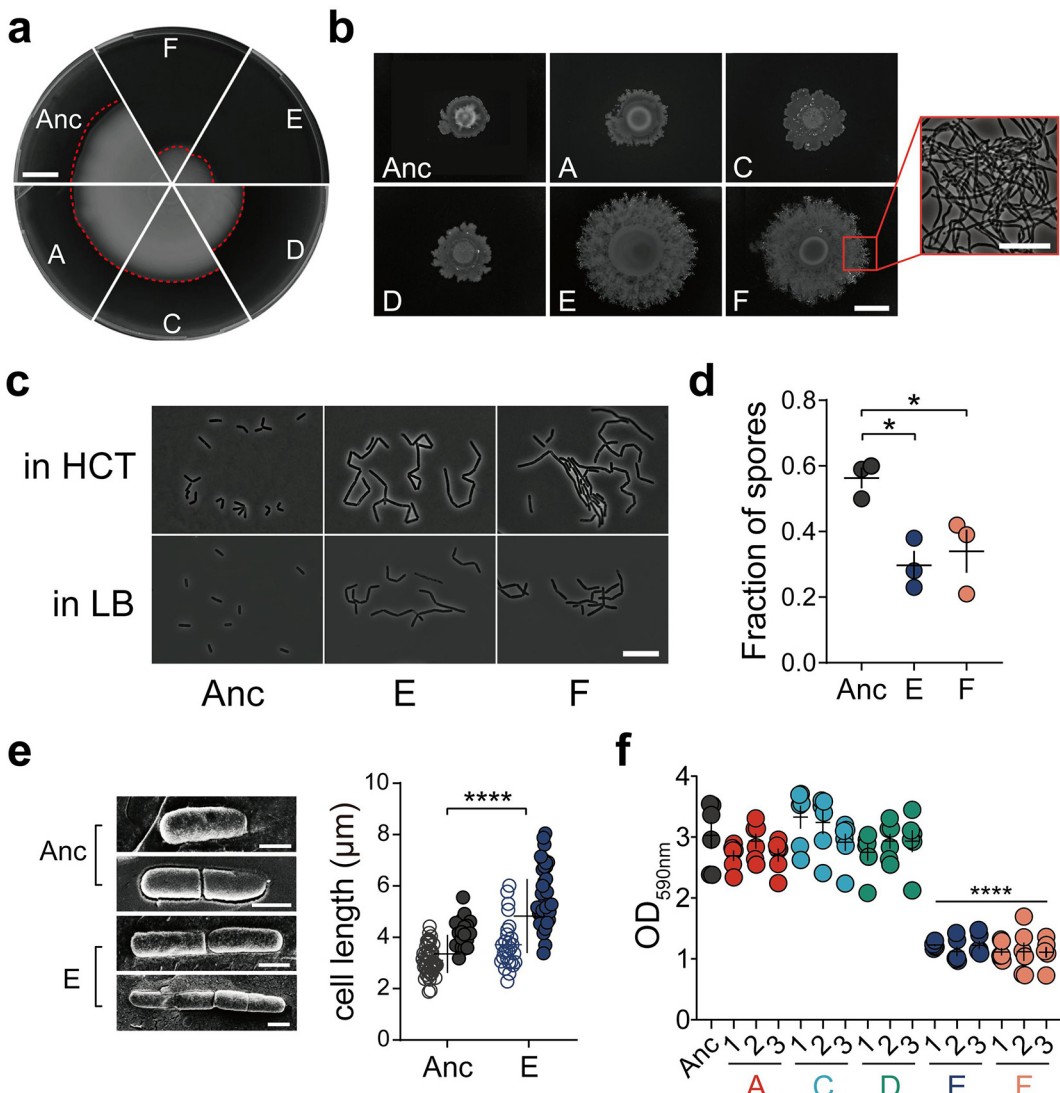

**FIG 2** Certain evolved lineages demonstrate elongated and sessile bacterial traits. (a) Swimming motility of Bt407 ancestor (Anc) and representative evolved isolates. Images are representative of three independent biological replicates. Bar, 1 cm. (b) Swarming radius of the Bt407 ancestor and evolved isolates. The framed area in panel F shows filamentous growth at the colony edge. Bars indicate 1 cm and 10 $\mu$m, respectively. Images are representative of three independent biological replicates. (c) Cell morphology of planktonic cultures at an $OD_{600}$ of 1 in HCT and LB media. Bar, 10 $\mu$m. Images are representative of three biologically independent bacterial cultures in HCT and LB media, respectively. (d) Sporulation of the ancestor strain and evolved lineages E and F ($n = 3$ independent biological samples per group). Vegetative cells and heat-resistant spores of three groups were counted after 48 h of incubation in HCT medium at 30°C. Error bars indicate the standard errors of the means. Statistically significant differences were examined using one-way analysis of variance (ANOVA) followed by Tukey's multiple comparisons (*, $P < 0.05$). (e) Cryo-SEM imaging of the Bt407 ancestor and evolved lineage E (left). Calculated length of single bacterial cells (open symbols) and chains (closed symbols) are shown on the right. The central values (horizontal lines) indicate the means ($n = 67$ for Bt407 ancestor and $n = 64$ for evolved lineage E), and the error bars represent standard deviations. Asterisks indicate significant differences (****, $P < 0.0001$; two-tailed $t$ test with Welch's corrections). Bars, 1 $\mu$m. (f) Surface-attached (submerged) biofilm formation, quantitated by crystal violet staining and subsequent solubilization by 70% ethanol. The central values (horizontal lines) represent the means ($n = 6$ biologically independent samples), and the error bars represent the standard errors of the means. Asterisks indicate significant differences between each group and the ancestor (****, $P < 0.0001$; two-tailed $t$ test with Welch's corrections).

elongation, which is either a requirement for or an indicator of swarming motility. In line with this, cell elongation was observed in the evolved E and F lineages (Fig. 2c).

Next, we tested sporulation of the evolved lineages. After 48 h, the Bt407 ancestor strain had sporulated efficiently in a medium favoring sporulation (HCT), while cells from lineages E and F were packed in aggregates with reduced levels of complete sporulation (Fig. 2d). When exposed to harsh conditions or competitors, sporulation

serves as a secondary defensive strategy in addition to being embedded in biofilm communities. Herein, we found that evolved isolates exhibited delayed or less frequent sporulation in a medium favoring sporulation (HCT), which may contribute to the efficacy of root colonization. At the cost of reducing sporulation efficiency, evolved biofilm aggregates might have a fitness advantage over swimming planktonic cells when competing for limited nutrients and/or colonizing plants.

Additionally, cryo-scanning electron microscopy (cryo-SEM) was used to examine bacterial ultrastructure and cell length (Fig. 2e), which further confirmed longer chains of unseparated cells of lineage E in addition to slightly increased single cell length, compared to the ancestor strain.

To further elucidate the sessile, nonmotile properties of certain evolved isolates, *in vitro* biofilm formation was assayed under different scenarios. Pellicle biofilm formed at the air-liquid interface generally requires motility as a pivotal factor during formation (57). As expected, pellicle formation in LBGM (58, 59) (LB with glycerol and $MnSO_4$) medium was greatly reduced in the evolved lineages E and F, providing additional evidence for pellicle formation being heavily dependent on swimming motility (Fig. S4A). Pellicles in evolved lineage A showed partial impairment, potentially indicating that pellicle formation could also be linked to other altered traits.

Next, the isolates were tested for submerged biofilm development at the plastic surface under slight shaking conditions. Both imaging and quantitative data demonstrated a significantly lower degree of submerged biofilm formation in lineages E and F, compared to the ancestor and other EE lineages (Fig. 2f and Fig. S4B). Meanwhile, the total growth of the ancestor and evolved strains was similar in the tested medium (Fig. S4C). Although pellicle formation and surface-attached biofilm formation are common techniques for screening biofilm formation in the laboratory, the list of genes required for these two biofilm models differs. The reduced biofilm formation of evolved isolates from lineages E and F in these two microenvironments implied spatial specialization of certain evolved strains, meaning that microbes may have tended to form aggregates in the medium instead of swimming to the air-liquid interface or the bottom of the microliter plates.

**Competitive advantage of evolved lineages during interaction with plants.** Following fundamental *in vitro* physiological tests, it is necessary to find why evolved isolates showed enhanced root colonization and whether they responded differently to plant materials compared with the ancestor. As proposed by Beauregard and colleagues (46), MSN medium supplemented with 0.5% cellobiose (MSNc) can be used to assess the plant-induced biofilm formation properties of *B. subtilis*. Cellobiose, a plant disaccharide resulting from hydrolyzed cellulose, is similar to that found on plants. However, MSNc medium alone did not induce biofilm formation as reported previously. Consistent with the literature on other *Bacilli*, we found that the ancestor strain could not form floating biofilms in this medium after 48 h but produced homogenous planktonic cultures displaying poor growth (Fig. 3A). In contrast, the evolved isolates formed various levels of floating aggregate-resembling biofilms. Dense structural aggregates (round shape and up to 10-mm diameter) with high compactness were observed for lineages E and F (Fig. 3A). Similarly, cellobiose also elevated the formation of aggregates in other evolved lineages, though varying in shape (stick-like) and size (up to 10 mm long) from those of lineages E and F. Cellobiose is a common component of the plant cell wall, which is present in soil and decaying plants, and can be utilized by some *B. cereus* strains as a carbon source (60). The elevated aggregation phenotype of evolved isolates suggests that an altered plant material-associated metabolic capacity and/or biofilm induction was favored under the EE regime.

To further explore the impact of plant polysaccharides on biofilm formation of the evolved isolates, pellicle formation was tested in LB medium supplemented with various plant polysaccharides. Among the polysaccharides previously investigated (46), xylan could induce dense and robust pellicle formation in the evolved isolates, whereas the pellicles of the ancestor remained thin and fragile (Fig. 3B). On the

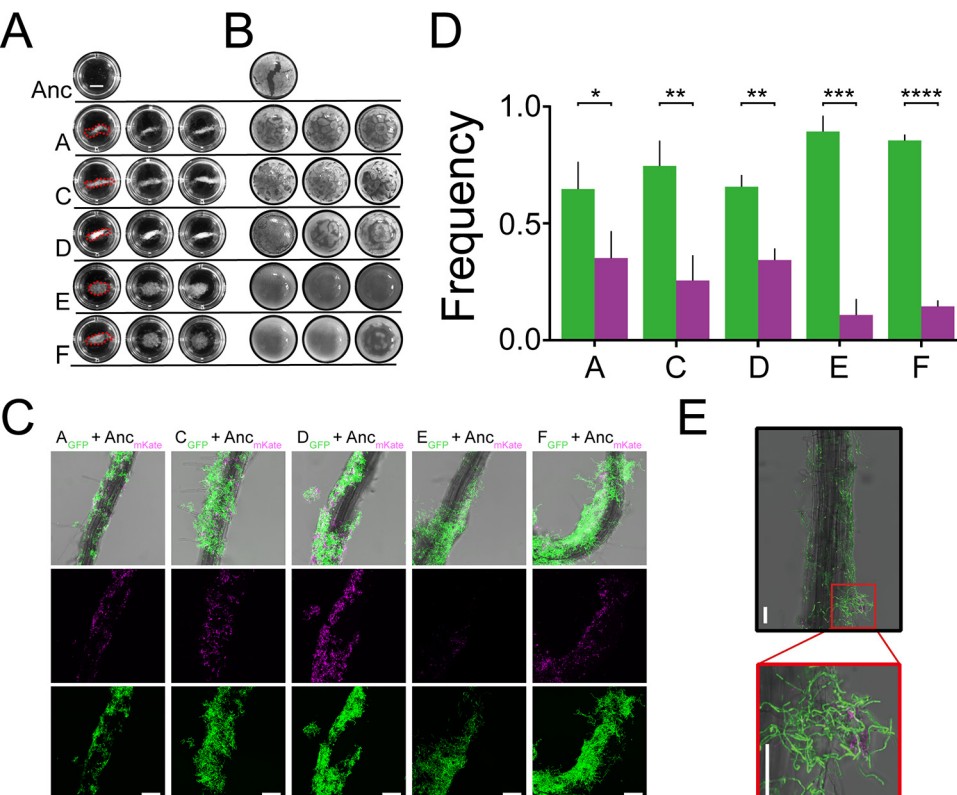

**FIG 3** Biofilm formation induced by plant polysaccharides is promoted in evolved lineages. (A) Images of biofilm aggregates in response to MSN medium supplemented with the plant polysaccharide cellobiose (0.5%) (MSNc). Biofilm analyses are representative of three biological replicates. Bar, 3 mm. (B) Pellicle formation of the ancestor and evolved isolates induced in LB medium supplemented with the plant polysaccharide xylan (0.5%). Panels A and B have the same magnification. (C) *A. thaliana* roots colonized by a 1:1 mixture of Bt407 ancestor (magenta) and evolved isolates (green) imaged by CLSM. The top row shows an overlay of fluorescence channels and the bright-field image. Images are representative of three independent seedlings. Bars, 50 $\mu$m. (D) Frequencies of each strain in the root-colonized biofilm. Bars represent the means ($n = 3$), and the error bars represent standard deviations (*, $P < 0.05$; **, $P < 0.01$; ***, $P < 0.001$; ****, $P < 0.0001$; two-tailed $t$ test with Welch's corrections). Color codes are as in panel C. (E) *B. thuringiensis* cells that carried either mKate (magenta, ancestor) or GFP (green, evolved isolate) reporter were cocultured with *A. thaliana* seedlings at a one-to-one ratio. After 48 h of incubation, root biofilms were visualized by CLSM. Bt407 ancestor and evolved isolate demonstrate different cell morphologies in root biofilms. Bars, 50 $\mu$m.

contrary, no robust biofilms were observed in samples cultivated either in LB medium without plant polysaccharide or in MSN medium with nonplant polysaccharide (glycerol) (Fig. S4D). This result is somewhat surprising, as no previous study has reported that LB medium, a rich medium, could induce pellicle formation with plant polysaccharide. Even in a nutrient-rich medium, evolved isolates could utilize plant polysaccharide as a signal and/or utilize it during biofilm assembly. Moreover, morphological differences were apparent among the evolved isolates; although the pellicles of lineages E and F were dense and compact, the absence of wrinkled structures suggests a distinct fundamental population behavior compared with the other evolved isolates from lineages A, C, and D. Therefore, plant polysaccharide and disaccharide could be a major factor, and possibly the only factor, causing the adaptation of evolved strains, and thus contribute to enhanced root colonization.

As the evolved lineages displayed altered differentiation properties and distinct biofilm formation in response to plant-derived polysaccharides and disaccharide, competitive colonization between the ancestor and evolved isolates was directly quantified on *A. thaliana* seedlings. Specifically, seedlings were seeded with combinations of ancestor and selected evolved isolates at a one-to-one ratio, carrying mKATE2 and green fluorescent protein (GFP) reporter, respectively. Within each evolved lineage, isolates

displaying the highest root colonization ability were selected for this assay. Concurrently, a control mixture of the ancestor labeled with two distinct fluorescent reporters was also assayed. Colonized biofilms were visualized by confocal laser scanning microscopy (CLSM) after incubation for 48 h with the seedlings. Unsurprisingly, evolved isolates had a competitive advantage over the ancestor in terms of root colonization (Fig. 3C and D). Importantly, the ancestor strain was most significantly outcompeted by evolved lineages E and F compared with the subtler competitive advantage of the other three lineages. Strikingly, cells in lineages E and F showed elongated and aggregated morphology on plant roots (Fig. 3E). Using a previously described method, the relative frequencies of each strain were quantified in root-colonized biofilms, based on total pixel volume (50). The frequency of evolved lineages E and F presented at a ratio above 85%, while other evolved lineages displayed frequencies of 60% to 80% (Fig. 3D). In contrast, the ratio of control ancestor mixtures remained approximately equal (Fig. S5A). Taken together, the competition assay results demonstrated that evolved strains were better at forming root colonization biofilms than the ancestor. In lineages E and F, elongated cell bundles attained the highest fitness, almost excluding colonization of the ancestor.

Intriguingly, when competition assays lasted for two cycles (4 days), very few vegetative ancestral bacteria were captured by CLSM (Fig. S5B). Conversely, the aggregates formed by evolved lineages E and F maintained the ability to colonize plants, showing delayed sporulation compared with the ancestor. This phenomenon was observed only for evolved lineages E and F. Ancestor-only cultures displayed a high degree of sporulation under these conditions. In view of the delayed sporulation observed for lineages E and F, these results may suggest that increased competitive recolonization of roots could be due to both increased aggregation/biofilm establishment and delayed sporulation on the root surface. We assumed that increased aggregate formation and delayed sporulation are associated traits in response to elevated root colonization. To test this hypothesis, we measured the sporulation kinetics of all strains in MSNc medium. In line with the results from CLSM imaging, cells of evolved lineages E and F exhibited the lowest sporulation efficacy at 96 h (Fig. 4A and B). Other evolved lineages also sporulated more slowly than the ancestor in MSNc medium, accompanied by an aggregating phenotype. Together, microscopic observations and sporulation frequency analyses suggested three distinct phenotypes for the ancestor and the evolved isolates (Fig. 4C). The aggregation phenotype typically provides bacteria or yeast with ecological benefits such as better nutrient uptake and protection from harsh environments (61, 62), and based on our results, possibly increased biofilm formation on the plant root surface.

**Plant-adapted E and F lineages display enhanced pathogenesis.** Although *B. thuringiensis* is well-known for its entomopathogenic traits, the strain used in our study is an acrystalliferous derivative of *B. thuringiensis* strain 407, and therefore less virulent via the oral route toward Cry-susceptible insects than its corresponding wild-type strain. Like *B. cereus*, Bt407 Cry⁻ could be considered a potential opportunistic pathogen that may cause food poisoning through the synthesis of pore-forming cytotoxins hemolysin BL (HBL), nonhemolytic enterotoxin (NHE), and cytotoxin K (CytK) (6), for which genes are present in this strain. Expression of these toxin genes is regulated by various transcriptional regulatory systems such as PlcR, ResDE, Fnr, and CcpA. Generally, these regulatory systems are synchronized with other bacterial behaviors, including motility, biofilm formation, and metabolism. In addition, differentiation to a swarmer cell, an attribute that was enhanced in the evolved lineages, has been associated with increased virulence properties in *B. cereus* (63). Thus, adapted Bt407 lineages might also exhibit reshaped virulence properties. First, hemolytic activity of the evolved isolates was assayed on brain heart infusion (BHI) agar medium containing sheep blood. Evolved lineages E and F displayed an increased hemolytic zone, a sign of more pronounced hemolysis, compared with the ancestor and other evolved lineages (Fig. 5A and B). Moreover, *in vivo* pathogenesis was tested against *Galleria mellonella*, a popular lepidopteran model, via injection into the hemolymph of insect larvae.

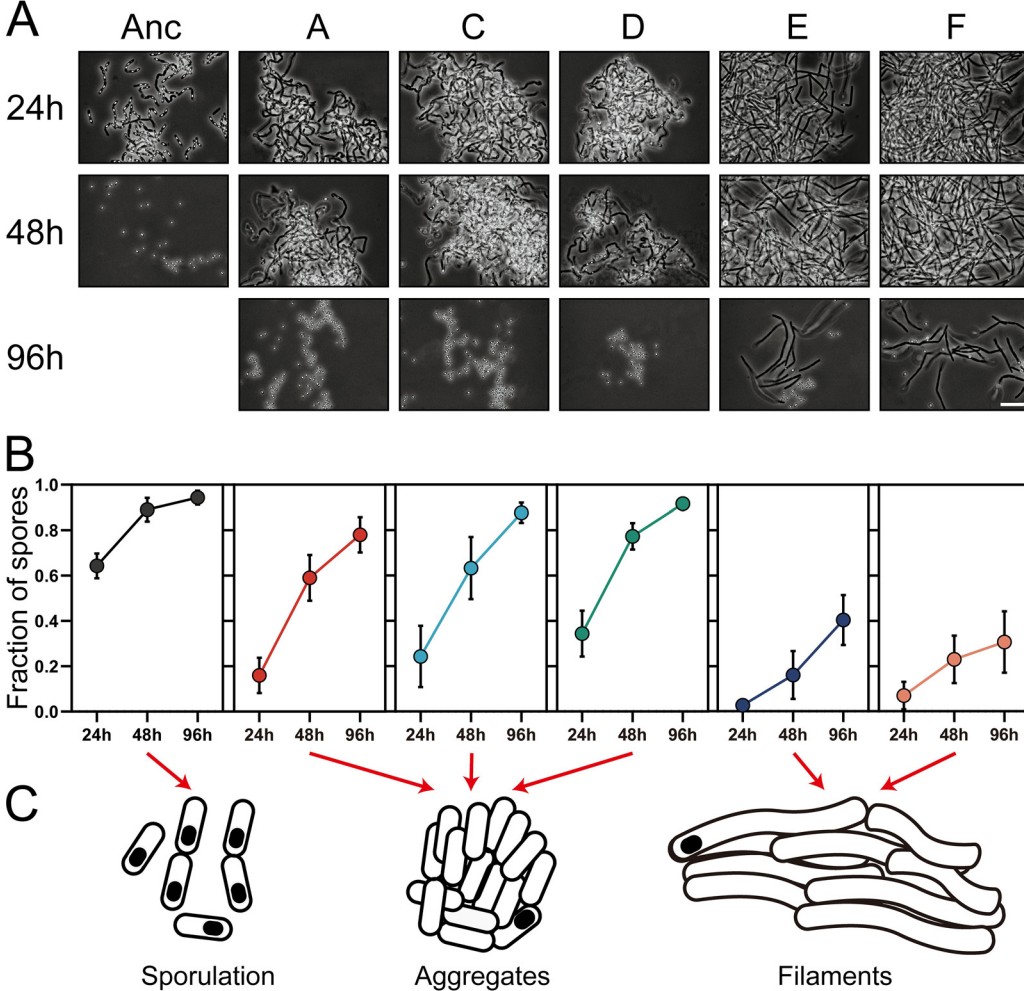

**FIG 4** Sporulation kinetics, aggregation response, and pathogenesis of the ancestor and evolved isolates. (A) Microscopic observations of strains in MSNc medium harvested at different time points. Images are representative of three independent cultures. Bar, 10 $\mu$m. (B) Sporulation percentage of strains in MSNc medium at three time points. Symbols represent the means of three replicative assays, and error bars represent the standard deviations. (C) Graphic representations of the three distinct cell morphologies observed.

Various concentrations of vegetative cells were employed to determine dose-response curves and 50% lethal dose ($LD_{50}$) values. Evolved isolates from lineages E and F exhibited a significant severalfold decrease in $LD_{50}$s compared to the ancestor strain (Fig. 5C), consistent with the hemolytic activity assay results. Taken together, both *in vitro* and *in vivo* assays revealed enhanced virulence properties of evolved lineages E and F. Alteration in phenotypes such as motility and biofilm formation might be closely linked to the pathogenicity of evolved strains. However, exactly how alternative non-host niches (the plant rhizosphere in this case) may affect the investment in cooperative virulence remains relatively unexplored. In *Pseudomonas aeruginosa*, it has been shown that the formation of aggregates confers enhanced virulence by selective up-regulation of quorum-sensing systems (64). Accordingly, autoaggregation of evolved lineages E and F may result in increased synthesis or secretion of effectors that promote virulence.

**Genomic analysis reveals key mutations related to evolutionary adaptation.** To identify whether evolved isolates harbored mutations responsible for the observed evolved phenotypic properties, whole-genome sequencing (WGS) analysis was carried out on three isolates from each evolved lineage and the ancestor. The analysis revealed a total of 58 mutations on the chromosome, most of which are nonsynonymous and

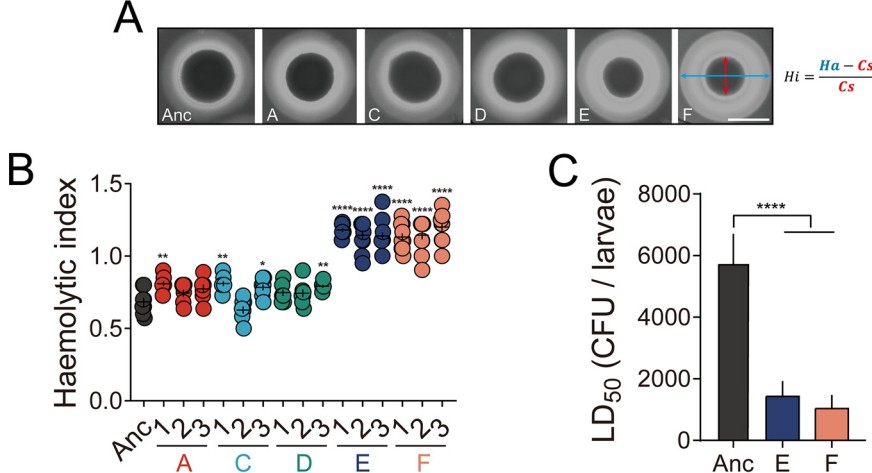

$$Hi = \frac{Ha - Cs}{Cs}$$

FIG 5 Pathogenesis of the ancestor and evolved isolates. (A) Representative images of hemolytic activity from ancestor and evolved isolate colonies. *Hi*, *Ha,* and *Cs* represent hemolytic indices, hemolytic areas, and colony sizes, respectively. Bar, 10 mm. (B) Hemolytic indices (as described in Materials and Methods). The central values (horizontal lines) represent the means ($n = 8$), while the error bars represent the standard errors of the means. Asterisks indicate significant differences between the values for each group and the ancestor (*, $P < 0.05$; **, $P < 0.01$; ****, $P < 0.0001$; two-tailed $t$ test with Welch's corrections). (C) $LD_{50}$ values of ancestor and evolved isolates based on mortality toward insect larvae (*Galleria mellonella*). Bars represent mean $LD_{50}$ values ($n = 3$), and error bars represent standard deviations. Asterisks indicate significant differences between each group and the ancestor (****, $P < 0.0001$; two-tailed $t$ test with Welch's corrections).

assigned to specific genes. Most mutations that were identified in all three isolates of a certain lineage were related to metabolic processes such as peptidoglycan and amino acid metabolism (see Data Set S1 in the supplemental material). Specifically, mutations in evolved lineages A, C, and D were present in genes related to peptidoglycan/cell wall metabolism. In contrast, lineages E and F harbored mutations in genes related to transcription and translation, which led to the hypothesis that certain genes (e.g., *rho*) might be responsible for different cell fate decisions in these evolved lineages.

Unlike *Escherichia coli*, the Rho-dependent transcription termination gene in *B. subtilis* is nonessential (65). Inactivation of *rho* alters the global transcriptome but allows robust growth in rich medium (66), and the dispensability of Rho in transcription provides plasticity in *B. subtilis* fitness. On the basis of these previous observations, we speculated that Rho-dependent termination might play a similar role in Bt407, and the nonsense mutation within the *rho* gene in lineages E and F might strongly influence diverse physiological processes. Notably, the observed mutation (*rho*^Glu54stop^) in lineages E and F presumably abrogates the full function of Rho since the stop codon is located before the RNA binding domain (Fig. 6a).

**Reintroduction of a nonsense mutation in *rho* imitates the key phenotypes of evolved lineages.** To dissect important genotype-phenotype relationships, the nonsense mutation in *rho* was targeted primarily, given its pivotal role in differentiation processes in *B. subtilis* (67), while other mutations will be examined in future studies. The *rho*^Glu54stop^ mutation (Fig. 6a) observed in lineages E and F was reintroduced into the Bt407 ancestor strain using homologous recombination. Similar to a previous study on *B. subtilis* (67), the nonsense mutation in *rho* impaired the swimming motility of Bt407, which partially imitated the severely reduced swimming radius of isolates from evolved lineage E (Fig. 6b and Fig. S6A). In line with the above results, the *rho*^Glu54stop^ strain was unable to form robust pellicles, displaying a flat and less wrinkled biofilm structure (Fig. 6b).

In a medium favoring sporulation (HCT) (68), cell chaining and bundle formation of the constructed *rho*^Glu54stop^ strain were comparable to isolates from evolved lineage E (Fig. S7A). Similarly, aggregate formation could also be observed in the *rho*^Glu54stop^ strain in response to cellobiose, although not as robust as the evolved E and F lineages

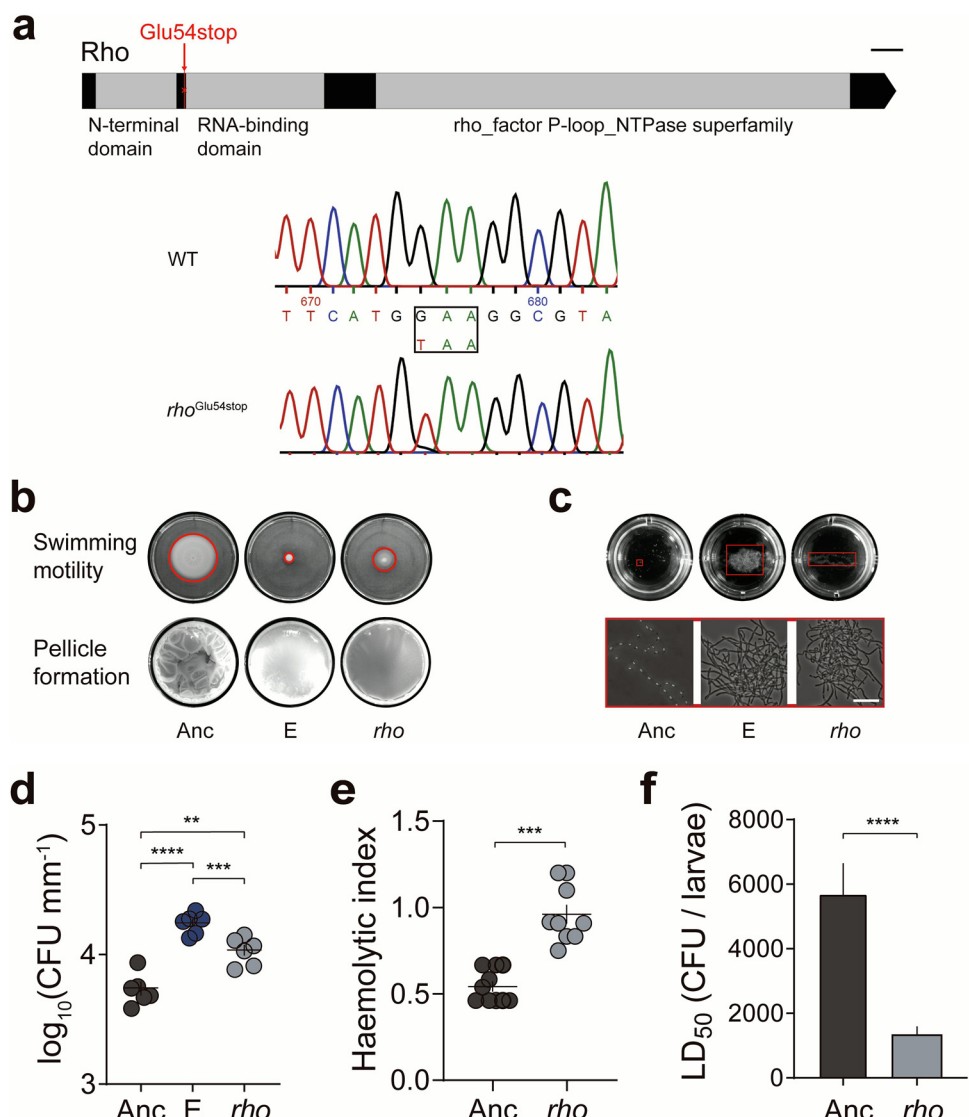

**FIG 6** The Rho transcriptional terminator plays a crucial role in cell fate decisions. (a) Schematic illustration of the location of the *rho*^Glu54stop^ mutation. The conserved motifs were identified through the NCBI conserved domain database (https://www.ncbi.nlm.nih.gov/Structure/cdd/wrpsb.cgi). WT, wild type. Bar, 50 nt. (b) Motility and biofilm formation in the *rho*^Glu54stop^ strain. Images are representatives of at least three independent replicates. (c) The plant disaccharide cellobiose promotes aggregate formation. Images are representative of at least three independent replicates. Bar, 10 μm. (d) Increased plant colonization of the E lineage isolate and the *rho*^Glu54stop^ strain. The central values (horizontal lines) represent the means (*n* = 6 biologically independent seedlings for all groups), and the error bars represent the standard errors of the means. For multiple comparisons, ordinary one-way ANOVA and Tukey comparison tests were employed (**, $P < 0.01$; ***, $P < 0.001$; ****, $P < 0.0001$). (e) Hemolytic activity is increased in the *rho*^Glu54stop^ strain. The central values (horizontal lines) represent the means (*n* = 10 biologically independent sampled cultures), and the error bars represent the standard errors of the means (***, $P < 0.001$; two-tailed *t* test with Welch's corrections). (F) *In vivo* toxicity of the *rho*^Glu54stop^ strain. The central values (horizontal lines) represent the means (*n* = 3), and the error bars represent standard deviations (****, $P < 0.0001$; two-tailed *t* test with Welch's corrections).

(Fig. 6c and Fig. S6B). Additionally, the higher root colonization ability of the *rho*^Glu54stop^ strain was confirmed, demonstrated by elevated CFU values compared with the ancestor (Fig. 6d). In agitated LB cultures, where ancestor cells were mostly dispersed, the *rho* mutant and the lineage E isolate both displayed an elongated cell morphology and reduced cell separation, determined by cryo-SEM imaging (Fig. S7B and S7C). Furthermore, inactivation of *rho* led to higher levels of hemolytic activity and insect larva toxicity compared with the ancestor, in line with the results from evolved lineages E and F, and suggesting that Rho directly or indirectly impacts the transcription

profile of certain genes responsible for pathogenesis (Fig. 6e and f). Combining these results, we hypothesized that the altered differentiation properties of evolved E and F lineages are primarily caused by the nonsense mutation in *rho*.

**The nonsense mutation in *rho* reshapes the transcriptional landscape of *B. thuringiensis*.** In order to reveal a potential molecular mechanism for how Rho affects the global gene expression profile of Bt407, we compared the transcriptomes of the ancestor strain, one isolate from evolved lineage E, and the *rho*^Glu54stop^ strain. Bacterial strains were cultured in LB medium until mid-log growth phase to ensure comparable growth with relatively low aggregate formation; therefore, the primary impact of mutation in *rho* is detected in the analysis (i.e., not the influence of altered plant colonization, aggregate formation, or delayed sporulation). Analysis of the transcriptomic data revealed that 377 and 270 genes were significantly (5% false discovery rate threshold) up- or downregulated, respectively, in the evolved isolate from lineage E compared with the ancestor (Data Set S2). In the case of the *rho*^Glu54stop^ strain, 523 and 378 genes were up- or downregulated, respectively. Most importantly, the evolved strain and the *rho*^Glu54stop^ strain shared a similar pattern of differentially expressed genes (DEGs) and Gene Ontology (GO) term categories (Fig. 7a to c).

As expected based on the phenotypic assays, hemolytic activity, and *in vivo* toxicity assays, genes with GO terms related to motility and chemotaxis were downregulated, while genes associated with pathogenesis were significantly upregulated in both the evolved isolate and the *rho*^Glu54stop^ strain compared with the ancestor (Fig. 7c and d and Data Set S3). In parallel with lineages E and F harboring mutations in genes related to the basic cellular processes of transcription and translation, downregulation of genes involved in translation, and to a lesser extent transcription, was observed in the transcriptome analysis (Fig. 7c). Adaptation of metabolic pathways related to plant polysaccharides was implied by the upregulation of genes responsible for the metabolism of various carbohydrates such as cellobiose, pyruvate, and galactose (Fig. 7d and Data Set S3). For instance, UDP-galactose, which is generated by the enzyme GalE, may serve as a substrate for the production of the extracellular polymeric substance (EPS) matrix, thereby helping microbes to form biofilms on plants. In *B. subtilis*, a *galE* mutant exhibited a decreased level of root colonization (46). Herein, upregulation of *galE* in the evolved isolate and the *rho*^Glu54stop^ strain may imply that the higher production rate of galactose contributes to enhanced root colonization in the evolved strains. Furthermore, RNA sequencing (RNA-Seq) data verified the adaptive cellobiose-related metabolic process in the evolved isolate and the *rho*^Glu54stop^ strain (Fig. 7d), concurrent with our hypothesis that adaptive metabolism of cellobiose successfully led to enhanced root colonization ability of the evolved isolates. Notably, the cellobiose-specific phosphotransferase system in *Klebsiella pneumoniae* contributes to biofilm formation (69). We reasoned that altered carbohydrate utilization may also influence sporulation of the *rho*^Glu54stop^ strain, for which only ~40% of cells were sporulating after 48 h in MSNc medium (Fig. S7D and S7E). Our transcriptome data revealed that genes related to carbohydrate metabolism were upregulated in evolved isolates exhibiting more extensive root colonization. Exploring the genomic features of bacterial adaptation to plants revealed that genomes of plant-associated bacteria encode significantly more carbohydrate metabolism functions than non-plant-associated bacterial genomes (70). In summary, we hypothesize that elevated carbohydrate metabolism and altered cellular physiology led to more pronounced aggregate formation by the *rho* mutant, providing higher fitness for root colonization.

## DISCUSSION

To the best of our knowledge, we presented the first laboratory experimental evolution of root-associated biofilms of *B. cereus* group species. First, we demonstrated that this experimental setup could select for improved colonizers, which is similar to a previous study using abiotic surfaces, a bead transfer model that is considered a robust method to screen for biofilm formers and study evolutionary processes (37). However, evolution on the root surface clearly differs from the adaptation pathway observed

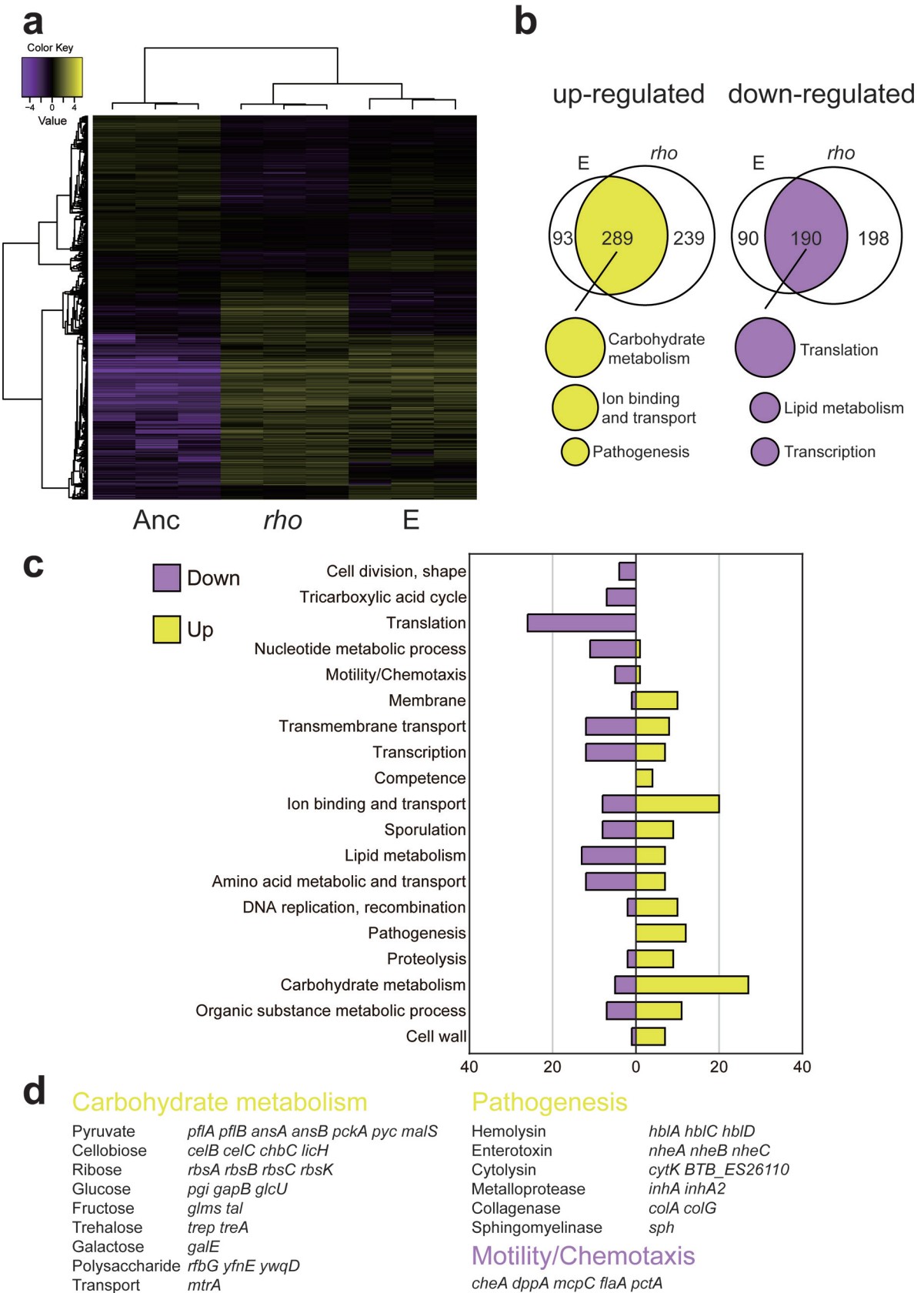

**FIG 7** An evolved isolate from lineage E and the *rho*^Glu54stop^ strain share similar differences of gene expression pattern compared with the ancestor. (a) Heatmap showing the relative expression levels of differentially expressed genes (DEGs) among the strains (*n* = 3

when *B. thuringiensis* is repetitively colonizing the abiotic surfaces of nylon beads (71). Evolved derivatives from the bead-adapted populations that displayed increased fitness compared to the ancestor carried an insertion sequence in the guanylyltransferase gene, *rfbM*. RfbM seems to influence cell surface properties of *B. thuringiensis*, and its mutation affects the cell adhesiveness to abiotic surfaces (71). On the contrary, mutations and transcriptional changes of plant-associated evolved derivatives were related to carbohydrate utilization, the effects of which were more pervasive. Furthermore, the mutational profiles of the evolved isolates in these two setups were distinct as well (71).

Under certain conditions, microbial cells develop biofilms, in which microbes thrive by forming tightly packed, multicellular aggregates, surviving harsh conditions (72). *Bacillus* spp. are recruited by small molecules secreted by plants and colonize plant roots by producing an exopolysaccharide matrix (41). Intriguingly, while the isolates from lineages E and F demonstrate improved recolonization of the seedlings, these strains display rather reduced biofilm formation *in vitro* or during simple plant attachment. As shown in Fig. S1 in the supplemental material, the biofilm biomass formed by E lineage is thicker than the seedling itself, which was easily underestimated by CFU because the biomass could be removed by shear force during processing of the samples. However, these tight cell interactions could benefit recolonization since the aggregation holds the dispersed cells together promoting rapid establishment of new biofilms (73). In the long term, the third dimension of cell aggregates of lineages E and F might have better access to nutrient resource (plant seedlings) and thus will be favored over single cells present in the ancestor strain. Competitive fitness of aggregates therefore might increase with time over single cells. Another advantage of multicellular aggregates in this scenario could be a shortened biofilm process, which typically requires multiple steps, including single cell attachment, biofilm growth, maturation, and dispersion (74). Overall, the benefits of multicellularity are broad and include the physical stickiness of cells that creates large and well-protected unit (75). In nature, bacteria tend to form multicellular aggregates in order to survive environmental stressors (76, 77). When cultivated in both soil and liquid soil extract, *B. cereus* employs a multicellular behavior to grow and translocate (78).

Bacterial aggregates have been observed on the surfaces of various plant species (79–81). Monier and colleagues have provided considerable evidence that cell aggregates are important traits for bacterial populations to survive on plant leaves (82, 83). In greenhouse experiments, the proportion of aggregated cells that were alive on plants was significantly higher than that of solitary cells (84). These multicellular structures are associated with putative nutrient sources released from plant veins or glandular trichomes. Besides, epiphytic bacteria in aggregates are surrounded by an exopolysaccharide matrix, which may provide a protective microenvironment. Aggregate formation could be one of the strategies how epiphytic bacterial populations survive on plants in nature.

Adaptation within the E and F lineages influenced various phenotypes, including sporulation and pathogenesis. Sporulation, a fundamental feature of *Bacilli*, is a powerful developmental program to live through harsh conditions. Despite the importance of *Bacilli* in plant protection, its sporulation on plant roots has remained uncharacterized. A recent study reported that *B. subtilis* in direct contact with plant seedlings sporulated slower than those surrounding plants (52). Interestingly, the evolved iso-

**FIG 7** Legend (Continued)
biologically replicates for each strain). TMM-normalized FPKM gene expression values were hierarchically clustered according to samples and genes. In the map, FPKM values are $\log_2$ transformed and median centered by gene. The color key gives the $\log_2$ value scale (negative and positive values represent gene expression below and above the median, respectively). (b) Venn diagrams of genes up- or downregulated in the evolved isolate from lineage E and the reconstructed *rho*^Glu54stop^ strain compared with the ancestor. The top functions of grouped DEGs are listed underneath the Venn diagrams. (c) Gene Ontology (GO) terms of the corresponding DEGs in the evolved strain and the *rho*^Glu54stop^ strain. (D) Representative functions and related DEGs. Yellow and magenta indicate up- and downregulated expression, respectively, in the evolved strain and the *rho*^Glu54stop^ strain relative to the ancestor.

lates from lineages E and F, especially the aggregated cells, sporulated slower than the ancestor. This delayed sporulation could be due either to altered global regulation of bacterial development or increased ability of the evolved isolates to consume plant-derived carbons causing delayed nutrient depletion. Whether the combination of enhanced root colonization and delayed sporulation is beneficial in nature will require confirmation in greenhouse experiments.

In this study, we observed a positive correlation between aggregated phenotypes and enhanced pathogenesis. For *B. cereus*, the expression of toxin genes can vary from strain to strain and largely affected by environmental conditions (3). EPS from biofilms of *B. cereus* contains enterotoxins such as HBL and NHE, along with other virulent factors (85). Biofilms serve as a protected environment in addition to being a niche for toxis accumulation. A recent study identified that a bifunctional protein, CalY, acts both as a virulence factor and a biofilm matrix component (86). Although biofilm formation might not be considered a direct virulent factor, it facilitates the survival of various pathogens in their hosts. The overproduction of these enterotoxins may facilitate competition against other organisms and could possibly enhance the fitness of E and F lineages, but this requires further validation in greenhouse experiments. Transcriptome data of the evolved isolate as well as the *rho* mutant revealed the upregulation of genes related to pathogenesis. These genes are under the control of the global transcriptional regulator PlcR (8); however, the transcript level of *plcR* remained unaltered. Further detailed experiments will be necessary to reveal how *rho* influences virulence factor production in the *B. cereus* group. Interestingly, CodY has been previously identified as an additional transcriptional regulator influencing toxin production in *B. cereus*. In addition to genes involved in virulence, CodY also regulates genes related to amino acid metabolism, biofilm formation, and energy production (87, 88), highlighting the general influence of global regulators on pathogenic potential of the *B. cereus* group.

Rho is essential in various bacterial species such as the enterobacterium *E. coli*. Nondeleterious mutations in *rho* within these species alter cellular fitness in the presence of various nutrients and antibiotics (89, 90). Unsurprisingly, mutations in *rho* have been found in numerous laboratory evolution studies (91–94). In *B. subtilis*, Rho-mediated transcriptome changes affect different cell differentiation programs such as cell motility, biofilm formation, sporulation, and antibiotic resistance (67, 95). We reasoned that in Bt407, *rho* mutation also leads to the multiple phenotypic changes such as altered motility, carbon metabolism, and toxicity. Additionally, although sporulation is influenced in the phenotypic assays, expression of sporulation genes remained unaltered under the conditions the samples were collected for transcriptome analysis.

In conclusion, an EE approach was designed to investigate the genotypic and phenotypic evolution of *B. thuringiensis* as biofilms associated with *A. thaliana*. Upon interaction with plant seedlings, evolved isolates accumulated mutations related to specific metabolic pathways, providing them with higher fitness during root colonization. Certain evolved lineages acquired a loss-of-function mutation, leading to disrupted Rho-dependent transcription termination, which eventually facilitated adaptive responses to a constantly changing environment, and efficient recolonization of plant seedlings. Fortuitously, loss of Rho function begat enhanced pathogenesis of these plant-adapted lineages, highlighting how pathogenesis may shift during environmental adaptation in the *B. cereus* group. Finally, evolved isolates form dense aggregates on the plant seedling or in response to plant polysaccharides, implying that plant-derived carbon may provide a driving force during laboratory evolution.

## MATERIALS AND METHODS

**Bacterial strains, media, and cultivation conditions.** Bacterial strains, plasmids, and primers used in this study are shown in Table 1. Bacterial strains were routinely cultured on lysogeny broth (LB-Lennox, Carl Roth; 10 g/liter tryptone, 5 g/liter yeast extract, and 5 g/liter NaCl) plates (1.5% agar) and stored at −80°C with 28% glycerol added. The following antibiotic concentrations were applied for cloning and plasmid maintenance: kanamycin (50 $\mu$g/ml), ampicillin (100 $\mu$g/ml), erythromycin (5 $\mu$g/ml), and tetracycline (10 $\mu$g/ml). X-Gal (5-bromo-4-chloro-3-indolyl-$\beta$-D-galactopyranoside) was used at 40 $\mu$g/ml.

**TABLE 1** Bacterial strains, plasmids, and oligonucleotides used in this study

| Bacterial strain, plasmid, or oligonucleotide | Description | Reference |
|---|---|---|
| **Bacterial strains** | | |
| *B. thuringiensis* 407 Cry⁻ (Bt407) | Acrystalliferous *B. thuringiensis* (Bt) type strain | 107 |
| Bt407 GFP | Bt407 transformed with pTB603 | This study |
| Bt407 mKate | Bt407 transformed with pTB604 | This study |
| Bt407 EvoA GFP | Bt407 evolved isolate from lineage A transformed with pTB603 | This study |
| Bt407 EvoC GFP | Bt407 evolved isolate from lineage C transformed with pTB603 | This study |
| Bt407 EvoDGFP | Bt407 evolved isolate from lineage D transformed with pTB603 | This study |
| Bt407 EvoE GFP | Bt407 evolved isolate from lineage E transformed with pTB603 | This study |
| Bt407 EvoF GFP | Bt407 evolved isolate from lineage F transformed with pTB603 | This study |
| Bt407 $rho^{Glu54stop}$ | Bt407 introduced a mutation of $rho^{Glu54stop}$ | This study |
| *E. coli* XL-1 Blue | *recA1 endA1 gyrA96 thi-1 hsdR17 supE44 relA1 lac* [F′ *proAB lacIqZ*ΔM15 Tn*10* (Tetʳ)] | 108 |
| **Plasmids** | | |
| pMAD | Shuttle vector, *bgaB*, *bla*, *ermC* (Ampʳ and Eryʳ) | 97 |
| pMAD (I-SceI) | pMAD containing I-SceI digestion site (Ampʳ and Eryʳ) | 98 |
| pBKJ223 | Vector expressing the I-SceI enzyme (Ampʳ and Tetʳ) | 96 |
| pTB603 | pNW33N with $P_{hyspank}$-GFP ($P_{hyspank}$-GFP was amplified from phyGFP [6]) | This study |
| pTB604 | pNW33N with $P_{hyspank}$-mKATE2 ($P_{hyspank}$-mKATE was amplified from phymKATE2 [6]) | This study |
| **Oligonucleotides** | | |
| oYL29 | Forward oligonucleotide to amplify the fragment with $rho^{Glu54stop}$ mutation CCG GAA TTC TTC AAT ATT ACG TGC CGC T | This study |
| oYL30 | Reverse oligonucleotide to amplify the fragment with $rho^{Glu54stop}$ mutation CGC GGA TCC TGG TGA ACT ATT ACG TGG T | This study |
| oYL41 | Forward oligonucleotide to test fragment integration in pMAD CGTCATATGGATCCGATATC | This study |
| oYL42 | Reverse oligonucleotide to test fragment integration in pMAD ATGGCATGCATCGATAGATC | This study |
| oYL45 | Forward oligonucleotide to test successful integration of pMAD with $rho^{Glu54stop}$ mutation into the chromosome of Bt GTTCCATATTTCCAGTTCC | This study |
| oYL46 | Reverese oligonucleotide to test successful integration of pMAD with $rho^{Glu54stop}$ mutation into the chromosome of Bt CCACAAAACGAAGCTGAA | This study |
| oYL49 | Forward oligonucleotide for sequencing the insert of multicloning sites of pMAD TCTATCGATGCATGCCAT | This study |
| oYL50 | Reverse oligonucleotide for sequencing the insert of multicloning sites of pMAD AGAATCATAATGGGGAAGG | This study |
| oTH1 | Forward oligonucleotide to amplify the $P_{hyspank}$-GFP/mKate2 cassette and clone into XbaI-EcoRI sites of pNW33N GCATCTAGAGTTGCTCGCGGGTAAATGTG | 50 |
| oTH2 | Reverse oligonucleotide to amplify the $P_{hyspank}$-GFP/mKate2 cassette and clone into XbaI-EcoRI sites of pNW33N CGAGAATTCATCCAGAAGCCTTGCATATC | 50 |

For all assays, to ensure proper disruption of aggregates in overnight cultures, samples were rigorously vortexed that ensured dispersion as validated by microscopy.

*A. thaliana* used in this study is belonging to ecotype Col-0. Plant seeds (around 30) were routinely surface sterilized in 2-ml centrifuge tubes using ethanol (70%, vol/vol) for 10 min. Subsequently, seeds were further sterilized in 1 ml of sodium hypochlorite (1%, vol/vol) and vigorously mixed using an orbital mixer for 15 min; thereafter, seeds were washed in sterile distilled water by repeated centrifugation and removal of supernatant for five times. After sterilization, seeds were suspended in 100 to 200 $\mu$l of water, subsequently planted on Murashige and Skoog basal medium (Sigma) 0.5% agar plates with 0.05% glucose. Plates were sealed with parafilm and incubated at 4°C for 3 days, followed by placing in a plant chamber with an angle of 65° (21°C, 16-h light and 20°C, 10-h dark). After 6 days, comparable seedlings ranging from 0.8 to 1.2 cm in length were selected for subsequent experiments.

**Construction of the fluorescent reporters and *rho* mutant.** For the construction of GFP and mKate reporters, $P_{hyspank}$-GFP and $P_{hyspank}$-mKATE2 regions were PCR amplified and cloned into the shuttle vector pNW33N. Derived vectors were designated pTB603 (GFP) and pTB604 (mKate), respectively.

A mutant with specific point nucleotide changes was created by homologous recombination using the markerless replacement method of Janes and Stibitz (96). First, the homologous fragments with desired mutations were PCR amplified and cloned into the modified pMAD shuttle vector (97), which contains additional I-SceI restriction site (98) (kindly provided by Toril Lindbäck). The plasmid carrying the intended mutation in the *rho* gene was verified by Sanger sequencing (Eurofins Genomics) and electroporated into Bt407 to obtain blue colonies on LB plates containing erythromycin and X-Gal. Integration of the vector into the chromosome was stimulated as described by Janes and Stibitz (96). Subsequently, pBKJ223 encoding I-SceI restriction enzyme was introduced, resulting in a double-stranded break at the chromosome and promoting a second recombination event (87). Finally, genomic

DNA was extracted from white colonies that have lost erythromycin resistance, and mutation was verified by Sanger sequencing of PCR fragments spanning the region of interest.

**EE of root colonization.** EE involving six parallel replicates was performed with an acrystalliferous derivative of *B. thuringiensis* 407 (Bt407 Cry⁻, referred as Bt407). The setup devised here was adapted from the concept of long-term biofilm EE on polystyrene beads (37), but instead of inert substrate, plant root was used for subsequent colonization by Bt407. Root colonization was carried out as described generally (46). Week-old *A. thaliana* seedlings (around 10 mm) were transferred into 300 $\mu$l of MSNg medium (5 mM potassium phosphate buffer, 100 mM morpholinepropanesulfonic acid [MOPS], 2 mM MgCl$_2$, 0.7 mM CaCl$_2$, 0.05 mM MnCl$_2$, 1 $\mu$M ZnCl$_2$, 2 $\mu$M thiamine, 0.2% NH$_4$Cl plus 0.05% glycerol as carbon source) in 48-well plates subsequently inoculated with Bt407 culture (optical density at 600 nm [OD$_{600}$] of 0.02). Then the plates were put on a laboratory shaker at 90 rpm in the climate chamber at 21°C. After 48 h, the seedlings with mature biofilm were washed gently to remove unattached cells in fresh MSNg medium, before they were transferred to a new plate hosting fresh medium and seedlings. When there were two seedlings floating in the medium, adhered cells emigrate from the old seedling to the new one. Every 48 h, the procedure was repeated to facilitate repetitive rounds of root colonization. The number of CFU per millimeter defined here as biofilm productivity was assessed regularly for the old seedlings. For this, the seedlings were washed, mixed with 100 $\mu$l of glass sand in 1 ml of 0.9% NaCl buffer, and vortexed vigorously for 5 min. The bacterial solution was diluted and plated for recording CFU. After 40 transfers, biofilm cells on the newly colonized plants were plated, and three single colonies were randomly selected and preserved for later analysis.

**Biofilm assays.** For pellicle formation assay, Bt407 cells were cultured overnight in LB medium at 37°C, subsequently diluted 1:100 in 3 ml of LB, and allowed to grow up to an OD$_{600}$ of <0.5, after which bacterial cultures were adjusted to an OD$_{600}$ of 0.3. Pellicle formation was assayed in 1 ml of LBGM or xylan-supplemented LB medium in a 24-well microtiter plate. LBGM was formulated as an efficient biofilm-inducing medium for *B. cereus* group bacteria and includes glycerol (1%) and MnSO$_4$ (100 $\mu$M) besides LB broth. For xylan-supplemented LB medium, xylan was used at a 0.5% (wt/vol) as plant-derived polysaccharide to induce pellicle formation. For each well of microtiter plates, 10 $\mu$l of adjusted cultures wwas added into the medium and incubated at 30°C for 48 h. Images were taken using a Panasonic DC-TZ90 camera.

Submerged biofilm formation was evaluated using a low nutrient medium (EPS medium described previously [99]). Overnight cultures of different strains were adjusted at an OD$_{600}$ of 0.2, 100 $\mu$l of which was added into 24-well plates containing 2 ml of EPS medium and incubated at 30°C at 50 rpm for 20 h. For quantitative analysis, total growth was measured at OD$_{620}$. After that, all planktonic cultures were removed, and plates were washed with 0.9% NaCl buffer. After air drying, adhered biofilms were stained with crystal violet solution (0.3%, 2 ml). After the solution was discarded from the wells, the unbound stain was removed with washing, the biofilm-bound crystal violet was solubilized with 70% ethanol, and its absorbance was detected at OD$_{590}$.

**Motility assays.** Soft agar plates (0.3% agar) were prepared for swimming assays. Overnight cultures of different strains were adjusted to OD$_{600}$ of 1, 5 $\mu$l was spotted on plates, and subsequently, the plates were incubated at 30°C. The swimming radius was measured after 20 h. Swarming assay was conducted on TrA medium (1% tryptone and 0.5% NaCl) containing 0.7% agar. Similar to swimming assay, overnight cultures of strains were adjusted to an OD$_{600}$ of 1, and then 5 $\mu$l was spotted on TrA agar medium, followed by incubation at 30°C. Images were obtained by an Axio Zoom V16 stereomicroscope (Carl Zeiss) equipped with a 0.5× Plan Apo objective and light-emitting diode (LED) cold-light sources.

**Sporulation assay.** Sporulation efficiency was evaluated in a defined sporulation medium HCT (100) and MSNc (46). Briefly, overnight cultures of strains were diluted in 10 ml if sporulation medium to obtain the exponentially growing cultures. Flasks were incubated at 30°C and 200 rpm, and at certain time points, samples were harvested and sonicated to disrupt cell aggregates (2 × 12 pulses of 1 s with 30% amplitude; Ultrasonic Processor VCX-130, Vibra-Cell; Sonics, Newtown, CT). Half of the sample was heated at 70°C for 20 min or left untreated, and both were serially diluted with 0.9% NaCl buffer and plated at LB agar plates. The sporulation efficiency was represented by the fraction of spores, calculated as the ratio of CFU in heat-treated compared to untreated samples.

**Hemolytic assays and insect larva experiments.** Hemolytic index was measured using 1.5% BHI agar plates supplemented with 5% defibrinated sheep blood (Thermo Scientific). After overnight incubation at 30°C, the hemolysis area and colony sizes were determined using ImageJ software, and the hemolytic index was calculated as previously described (101). For virulence experiment, wax moth larvae (*Galleria mellonella* larvae, obtained from Creep4you A/S) were used. Four dilutions (approximately 10$^3$ to 10$^6$ of cells) of overnight cultures were prepared, corresponding CFU was determined, and 10 $\mu$l of each dilution was injected into the hindmost left prolegs of larvae using 10-$\mu$l Hamilton syringes, while the control larvae were treated with 0.9% NaCl buffer. The experiment was repeated three times with a minimum of 20 larvae in each group. After incubation at 37°C for 24 h, the mortality of each group was recorded. The LD$_{50}$ of each strain was evaluated based on mortality data and calculated by the probit regression analysis employing IBM SPSS Statistics 20.

**Microscopy.** For bright-field images of pellicles and colonies, Axio Zoom V16 stereomicroscope (Carl Zeiss) was used, equipped with a Zeiss CL 9000 LED light source, a PlanApo Z 0.5× objective, and AxioCam MRm monochrome camera (Carl Zeiss).

Confocal laser scanning microscopy imaging was performed as described previously (50). For root-attached biofilms, colonized plants were washed twice with sterilized double-distilled water (ddH$_2$O) and placed onto glass slides. Images were obtained using a 63×/1.4 oil objective. Fluorescent reporter excitation was performed with the argon laser at 488 nm, and the emitted fluorescence was recorded at

484 to 536 nm and 567 to 654 nm for GFP and mKate, respectively. Z-stack series were performed with 1-$\mu$m steps, and stacks were merged using ImageJ software.

For cryo-electron microscopy imaging, bacteria were grown overnight at 30°C, $OD_{600}$ was adjusted to 1 using NaCl buffer (0.9%), 1-ml portions of the adjusted cultures were centrifuged washed three times using NaCl buffer (0.9%), and the resulting cell pellets were subjected to cryo-SEM analysis. Cryo-fixation was performed via high-pressure freezing (HPF) before cryo-SEM. The HPF was performed by the HPF instrument HPM100 (Leica Microsystems) in standard conditions. After freezing, the sample was mounted into a cryo-sample holder in liquid nitrogen attached to a Leica VCT100 cryo transfer arm (Leica Microsystems) and transferred into a Leica MED020 freeze fracture and coating system. After the freeze fracturing in the MED20 cryo-preparation chamber (Leica Microsystems), the biofilm freezing samples were sublimated at $-90$°C for about 90 s and coated with 6 nm of a C/Pt alloy. The biofilm samples were then transferred by the VCT 100 shuttle (Leica Microsystems) into a Quanta 3D FEG cryo-SEM (Thermo Fisher Scientific) and observed at 2 kV at $-160$°C. The cryo-SEM imaging was performed at the Core Facility for Integrated Microscopy, Copenhagen University, Denmark. The image analysis and measurement of the bacteria length were performed by using ImageJ software. Cells in chains of two or more individuals were enumerated and categorized as "chains," whereas one individual was categorized as a single cell.

**Genome resequencing.** Genomic DNA was extracted from overnight cultures using EURx Bacterial and Yeast Genomic DNA kit. Paired-end libraries were prepared using the NEBNext Ultra II DNA Library Prep kit for Illumina. Paired-end fragment reads were generated on an Illumina NextSeq sequencer using TG NextSeq 500/550 High Output kit v2 (300 cycles). Primary data analysis (base calling) was carried out with "bcl2fastq" software (v2.17.1.14; Illumina). All further analysis steps were done in CLC Genomics Workbench Tool 9.5.1. Reads were quality trimmed using an error probability of 0.05 (Q13) as the threshold. In addition, the first 10 bases of each read were removed. Reads that displayed ≥80% similarity to the reference over ≥80% of their read lengths were used in the mapping. Nonspecific reads were randomly placed to one of their possible genomic locations. Quality-based single nucleotide polymorphism (SNP) and small in/del variant calling was carried out requiring ≥8× read coverage with ≥25% variant frequency. Only variants supported by good-quality bases ($Q \geq 20$) were considered and only when they were supported by evidence from both DNA strands in comparison to the *B. thuringiensis* Bt407 genome (GenBank accession number CP003889.1). Identified mutations in each strain are listed in Data Set S1 in the supplemental material. Raw sequencing data have been deposited to the NCBI Sequence Read Archive (SRA) database under BioProject accession number PRJNA673616.

**RNA extraction and transcriptome analysis.** Overnight-grown strains were diluted to an $OD_{600}$ of 1.0 in LB medium and grown to late-log phase, and cultures were collected by centrifugation and flash frozen in liquid nitrogen. Total RNA was extracted by combining phenol-chloroform-isopropanol treatment using the High Pure RNA isolation kit (Roche, Germany) (102). The concentration and quality of extracted RNA were examined by Nanodrop and Agilent RNA 6000 Nano Chips. Ribo-off rRNA Depletion kit (Bacteria) (Vazyme Biotech) was used to deplete rRNA. *In vitro* fragment libraries were prepared using the Illumina Truseq RNA Library Prep kit v2, library qualities were controlled by Agilent Tapestation 2200. Paired-end fragment reads were generated on an Illumina NextSeq sequencer using TG NextSeq 500/550 High Output kit v2 (300 cycles). Reads were quality trimmed using an error probability of 0.05 (Q13) as the threshold.

Transcriptome reads were first preprocessed with cutadapt 1.1 (https://cutadapt.readthedocs.org/) to remove remains of Illumina adapters (error rate of 10% and overlap of ≥10 nucleotides [nt]; "-e 0.10 -O 10"), then quality trimmed using prinseq-lite (103) to remove the first 10 bases, trim from both ends with a quality threshold of ≥20, and retain reads with a mean quality of ≥28 and a length of ≥30 nt ("-trim_left 10 -trim_qual_right 20 -trim_qual_left 20 -min_qual_mean 28 -min_len 30"). Read pairs for which both mates passed the quality control (QC) checks were recovered using cmpfastq.pl (https://toolshed.g2.bx.psu.edu/repos/amol/cmpfastq), were generated, followed by differential gene expression analyses using Trinity 2.10.0 with RSEM 1.3.3 to estimate gene/transcript abundances and DESeq2 v1.22.2 run in R 3.5.3 (http://www.R-project.org/) for statistical testing (104). In RSEM, the bowtie2 (105) v2.4.1 aligner was employed to map (with a maximum insert size of 1,000; "–max_ins_size 1000") the read pairs onto the annotated protein-coding (coding DNA sequence [CDS]) transcript sequences of Bt407 (GenBank: GCF_000306745.1, file "cds_from_genomic.fna"). Genes that have *P* values adjusted for multiple testing (adjusted P value [$P_{adj}$]; corresponding to the false discovery rate [FDR]) ≤ 5% were considered DEGs and included for further analysis (Data Set S2). A heatmap with hierarchical tree clustering of genes and replicate samples according to gene expression profiles was computed using the "analyze_diff_expr.pl" utility script in Trinity. For that, gene/transcript abundances were converted (with the "run_TMM_normalization_write_FPKM_matrix.pl" utility) to TMM-normalized (trimmed mean of M values) FPKM (fragment per kilobase per million mapped reads) expression values. The Pearson correlation coefficient was chosen as the distance metric, and average linkage was chosen as the clustering method (options "–sample_dist sample_cor –sample_cor pearson –sample_clust average"). A $\log_2$ cutoff of 0 and a *P* value cutoff of 0.05 were set ("-C 0 P 0.05") to select DEGs (other options included "–min_rowSums 10 –min_colSums 0"). Functional analysis of DEGs was carried out using the Genome2D webserver (106) and GSEA-Pro v3.0 webserver (http://gseapro.molgenrug.nl/), combining FACoP (http://facop.molgenrug.nl/) and database (UniProt and SubtiWiki) information (Data Set S3).

Raw sequencing data have been deposited to the NCBI Sequence Read Archive (SRA) database under BioProject accession number PRJNA673582.

**Statistical analysis.** Unless indicated otherwise, all experiments were performed with at least three biological replicates. Statistical analysis of bacterial trait comparison between evolved isolates and the

ancestor (e.g., root colonization assays) was analyzed and illustrated using GraphPad Prism 8. Specifically, the statistical differences were calculated using Student's *t* test with Welch's correction (assuming unequal standard deviations [SD]). For comparison across multiple groups (e.g., comparison among the ancestor, evolved strain E, and constructed *rho* mutant), ordinary one-way analysis of variance (ANOVA) analysis and Tukey tests were employed.

**Data availability.** Further information and requests for resources and reagents should be directed to and will be fulfilled by the lead contact, Ákos T. Kovács (atkovacs@dtu.dk). Identified mutations, normalized transcriptome data are available in Data Sets S1 to S3, while raw sequencing reads have been submitted to NCBI Sequence Read Archive (SRA) database under BioProject accession numbers PRJNA673616 and PRJNA673582 (see above).

## SUPPLEMENTAL MATERIAL

Supplemental material is available online only.
**DATA SET S1**, XLSX file, 0.02 MB.
**DATA SET S2**, XLSX file, 0.4 MB.
**DATA SET S3**, XLSX file, 0.1 MB.
**FIG S1**, TIF file, 0.4 MB.
**FIG S2**, TIF file, 0.9 MB.
**FIG S3**, TIF file, 1.4 MB.
**FIG S4**, TIF file, 2 MB.
**FIG S5**, TIF file, 1 MB.
**FIG S6**, TIF file, 1.3 MB.
**FIG S7**, TIF file, 0.8 MB.

## ACKNOWLEDGMENTS

Y.L. was supported by a Chinese Scholarship Council fellowship. G.M. was supported by the Lendület-Program of the Hungarian Academy of Sciences (LP2020-5/2020).

All authors contributed to, and approved, the final version of the paper. Y.L. and Á.T.K. conceived the project. Y.L. performed the experiments and analyzed the data. M.A. and J.K.Y.P. constructed the *rho* mutant. M.E.-R. performed SEM. Y.L. and N.T. performed the RNA-Seq analysis. O.A.Ø. and G.M. provided resources and analysis methods. Á.T.K. supervised the project. Y.L. and Á.T.K. wrote the manuscript with input from all authors.

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
