## [Reviewer comments · mSystems]

Adaptation of *Bacillus thuringiensis* to plant colonization affects differentiation and toxicity

Yicen Lin, Monica Alstrup, Janet Ka Yan Pang, Gergely Maróti, Mériem Er-Rafik, Nicolas Tourasse, Ole Andreas Økstad, and Ákos T Kovács

Corresponding Author(s): Ákos T Kovács, Technical University of Denmark

Review Timeline:

Submission Date:	July 2, 2021
Editorial Decision:	August 13, 2021
Revision Received:	September 9, 2021
Editorial Decision:	September 26, 2021
Revision Received:	September 27, 2021
Accepted:	September 27, 2021

Editor: Elizabeth Shank

Reviewer(s): The reviewers have opted to remain anonymous.

Transaction Report:

DOI: <https://doi.org/10.1128/mSystems.00864-21>

August 13, 2021

Prof. Ákos T Kovács
Technical University of Denmark
Department of Biotechnology and Biomedicine
Søltofts Plads 221
Kgs Lyngby 2800
Denmark

Re: mSystems00864-21 (Adaptation of *Bacillus thuringiensis* to plant colonization affects differentiation and toxicity)

Dear Prof. Ákos T Kovács:

Thank you for submitting your manuscript to mSystems. We have now completed our review of your manuscript and the two reviewers were overall very enthusiastic about your work. However, one major concern raised by one of the reviewers (which I share) is that the study as currently assembled does not directly demonstrate that the adaptation observed is plant-specific. This concern (along with the others noted below and attached) must be addressed before the manuscript can be considered for publication in mSystems. As a minor note, the manuscript currently contains more than ten supplemental files, which would need to be consolidated during any resubmission.

Preparing Revision Guidelines

For complete guidelines on revision requirements for your article type, please see the journal Article Types requirement at <https://journals.asm.org/journal/mSystems/article-types>. **Submissions of a paper that does not conform to mSystems guidelines will delay acceptance of your manuscript.**

Sincerely,

Elizabeth Shank

Editor, mSystems

Journals Department
Reviewer comments:

Reviewer #1 (Comments for the Author):

With great interest I've read this manuscript by Lin et al. It is very well written and structured and therefore easy to follow. The figures and data presentation are excellent. I would like to congratulate the authors with this nice study and have only a few small comments/suggestions to further enhance reading:

- In the abstract, toxicity is called 'unrelated' to pathogenesis as these changes in pathogenicity are supposed to be unrelated to the enhanced virulence towards insects in view of the EE on the plant (I suppose). This makes sense to me as well, but the sentence in the abstract could benefit from some more detail.
- It would be nice to explain why the Cry- strain was used. What was the reason?
- L95: add 'of' in front of 'its ecology'
- L161: not clear why there is a lower case, bold 'c' in the S7 figure legend
- Why are no statistics included in the Figure 3D panel?
- L303: Not entirely clear why panel D is included with S4. Wouldn't it be better placed as inset with main Figure 3?
- L335: It would be nice to include some more details on the representative images of the hemolytic activity. Given this is a paper of general interest, especially also to plant scientists, it would be beneficial to explain a little bit more for non-experts. What are we looking at? NB: In general this would be beneficial for some other detailed phenotypes such as the pellicle formation, too.
- L359: Maybe it would be better to split the first (A-C) and second halves (D-F) of figure 4? Easier interpretation and integration with the separate sections in the manuscript.
- L409: Lineage E isn't shown in Fig S7 (it would be beneficial though; for interpretation as well as in line with the rest of the manuscript)?
- L682: It would be nice to include methodological and material (like plasmids) details on how the fluorescent reporters were created.
- L760: The indicated bioproject only contains data of the 16 genomes; please also add the transcriptome data (or the relevant bioproject number).

Adaptation of *Bacillus thuringiensis* to plant colonization affects differentiation and toxicity.

Much research has been performed in recent years to characterize the composition and diversity of the plant microbiome, especially in the model plant *Arabidopsis thaliana*. However, the molecular mechanism of bacterial colonization and adaptation to the plant environment is still limited. In this paper, the authors utilized experimental evolution to explore *B. thuringiensis* to the root environment. The authors characterized two evolved lineages and demonstrated that they harbor mutations in the transcription termination factor Rho leading to a transcriptional rearrangement. The authors further suggested that this mutation makes these isolates better root colonizers by promoting multicellular behaviors, like aggregation and swarming, and better utilization of plant sugars.

Major issues:

The paper gives an excellent demonstration of the utility of experimental evolution for understanding mechanisms of root colonization.

1. However, the experimental setting was devised for evolving increased biofilm formation and dispersal on polystyrene beads. It is not clear that the plant plays any role in this adaptation as "a plant," not just as a substrate for biofilm attachment. In addition, the condition of the experiment including shaking, and switching from plant to plant in each cycle, seem very artificial regarding root colonization in more natural settings. The authors need to strengthen the notion that the bacteria adapted to plant colonization, as this is the paper's main point. This may be done by monitoring root colonization of the evolved population in other conditions like agar plates and soil.

2. The only proof that bacteria adapted to plant colonization is that the evolved bacteria respond differently to plant material compared with the ancestor. Two controls are lacking to prove this point 1) biofilm formation in LB media without Xylan 2) Biofilm formation on MSN medium with non-plant sugars (Beauregard P, PNAS 2013).

3. The differences in biofilm formation shown in figures 3A-3B and 5C need to be quantified. These are major results for the paper, and the pictures alone are not compelling enough

Minor issues:

1. The legend of figure 1A says: "plates were agitated at 50 rpm," while in the methods section, it is written that "plates were put on a laboratory shaker at 90 rpm".
2. There is no explanation of how the authors differentiate single cells from chains (figure 2E and S7C). (membrane staining?)
3. The title of figure 6 is not clear "**An evolved isolate from lineage E and the *rhoGlu54stop* strain share similar gene expression patterns with the ancestor.**" The title suggests that Lineage E and *rhoGlu54stop* share the expression pattern **with** the ancestor. The truth is that they both share the expression pattern **differences compared** to the ancestor.

Reviewer #1 (Comments for the Author):

With great interest I've read this manuscript by Lin et al. It is very well written and structured and therefore easy to follow. The figures and data presentation are excellent. I would like to congratulate the authors with this nice study and have only a few small comments/suggestions to further enhance reading:

- In the abstract, toxicity is called 'unrelated' to pathogenesis as these changes in pathogenicity are supposed to be unrelated to the enhanced virulence towards insects in view of the EE on the plant (I suppose). This makes sense to me as well, but the sentence in the abstract could benefit from some more detail.

We agree with the Reviewer that toxicity and pathogenesis might be confusing in the abstract, therefore we decided to only state pathogenesis, as this has been tested in our experiments.

- It would be nice to explain why the Cry- strain was used. What was the reason?

A: Thank you for pointing this out. Lereclus et al. (1) firstly obtained Bt407 Cry- strain by shifting the Cry+ bacterial culture to 42 °C. The benefits of Cry- strain were significant such as stable heterologous expression and easy molecular modification (1, 2). Since then, this acrySTALLIFEROUS derivative has been widely used as a model strain in many studies (3–8). The informative background is the first reason we used Cry- strain.

Another important reason is that considering the ecological importance of *B. cereus* group species, acrySTALLIFEROUS bt407 will be an ideal agent that resembles *B. cereus*, thus providing broader impact of this study.

We have now referred to this information in the article.

- L95: add 'of' in front of 'its ecology'

A: Thanks for noticing. Corrected.

- L161: not clear why there is a lower case, bold 'c' in the S7 figure legend

A: Corrected.

- Why are no statistics included in the Figure 3D panel?

A: Thanks for the comment. We have added the proper statistics into the Fig. 3D panel and revised the figure.

- L303: Not entirely clear why panel D is included with S4. Wouldn't it be better placed as inset with main Figure 3?

A: Thanks for the suggestion. We have moved the Fig. S4D to Fig. 3 as panel E. It did help we improve the flow of the manuscript and make the paper easier to follow.

- L335: It would be nice to include some more details on the representative images of the hemolytic activity. Given this is a paper of general interest, especially also to plant scientists, it would be beneficial to explain a little bit more for non-experts. What are we looking at? NB: In general this would be beneficial for some other detailed phenotypes such as the pellicle formation, too.

A: Thanks for the suggestions. We have added a description beside the representative images of the hemolytic activity results, which should enhance the understanding of how hemolytic index was calculated. However, only qualitative analysis exists for pellicle formation and used in the Bacillus field in the last 2 decades, which describes the structural complexity of pellicles.

- L359: Maybe it would be better to split the first (A-C) and second halves (D-F) of figure 4? Easier interpretation and integration with the separate sections in the manuscript.

A: It's a very good comment and we have split the figure into two as suggested.

- L409: Lineage E isn't shown in Fig S7 (it would be beneficial though; for interpretation as well as in line with the rest of the manuscript)?

A: Thanks for pointing this issue. While it would be intuitive to put the results of anc, E and rho together, the results of lineage E in Fig. S7 have been introduced previously. To avoid repeatedly reporting the exact same image, only comparison of rho mutant and anc was included in that supplementary image.

- L682: It would be nice to include methodological and material (like plasmids) details on how the fluorescent reporters were created.

A: Thanks for the comment. The material of creating the fluorescent reporters has been now included in Table 1 and description of the methodology was added in Materials and methods.

- L760: The indicated bioproject only contains data of the 16 genomes; please also add the transcriptome data (or the relevant bioproject number).

A: The BioProject accession number for the transcriptome has been now adjusted and correctly displaying the right link: PRJNA673582. Thank you for noticing this!

Reviewer #2 (Comments for the Author):

Much research has been performed in recent years to characterize the composition and diversity of the plant microbiome, especially in the model plant *Arabidopsis thaliana*. However, the molecular mechanism of bacterial colonization and adaptation to the plant environment is still limited. In this paper, the authors utilized experimental evolution to explore *B. thuringiensis* to the root environment. The authors characterized two evolved lineages and demonstrated that they harbor mutations in the transcription termination factor Rho leading to a transcriptional rearrangement. The authors further suggested that this mutation makes these isolates better root colonizers by promoting multicellular behaviors, like aggregation and swarming, and better utilization of plant sugars.

Major issues:

The paper gives an excellent demonstration of the utility of experimental evolution for understanding mechanisms of root colonization.

1. However, the experimental setting was devised for evolving increased biofilm formation and dispersal on polystyrene beads. It is not clear that the plant plays any role in this adaptation as "a plant," not just as a substrate for biofilm attachment. In addition, the condition of the experiment including shaking, and switching from plant to plant in each cycle, seem very artificial regarding root colonization in more natural settings. The authors need to strengthen the notion that the bacteria adapted to plant colonization, as this is the paper's main point. This may be done by monitoring root colonization of the evolved population in other conditions like agar plates and soil.

A: Thank you for pointing this issue. The EE setup in this study was indeed a modified version of the well know bead model that was previously used to test adaptation on abiotic surfaces. In the meantime, we have also experimentally evolved the bacteria in the bead system to test adaptation to abiotic biofilm formation. The work has been now deposited to the bioRxiv (9).

We reasoned that the microbe has distinct evolutionary patterns in these two EE setups. For example, in the bead system, evolved variants with the highest fitness mutated in a guanylyltransferase gene related to the cell surface properties. The mutation has very specific effects on the adhesiveness of the cells to the abiotic surfaces. On the contrary, the mutation in plant-associated evolved bacteria was more related to the carbohydrate utilization, the effects of which were more pervasive. Furthermore, the mutational profiles of the microbes in these two setups were distinct as well. In plant-associated evolved variants, 70 SNPs were identified. For bead-associated variants, the number was 27. Importantly, there were few overlapping mutations among them in both setups. Taking together, we believed this evolutionary pattern was specifically shaped by plant-associated biofilms. Thus, comparison of the two setups, biotic (plant) and abiotic (bead) and the derived conclusions in the two manuscripts clearly demonstrate distinct adaptation patterns.

2. The only proof that bacteria adapted to plant colonization is that the evolved bacteria respond differently to plant material compared with the ancestor. Two controls are lacking to prove this point
1) biofilm formation in LB media without Xylan 2) Biofilm formation on MSN medium with non-plant sugars (Beauregard P, PNAS 2013).

A: Thank you for the very constructive comment. We agree that these two controls would be necessary in the results. Not surprisingly, in LB and MSN plus non-plant sugar (glycerol in this case), Bt407 could not form robust biofilms. The representative images were included in the supplementary Fig. S4.

3. The differences in biofilm formation shown in figures 3A-3B and 5C need to be quantified. These are major results for the paper, and the pictures alone are not compelling enough.

A: Thank you for the comment. Generally, pellicle development provides only a qualitative measure for biofilm formation. It is well accepted approach in the Bacillus field for the last two decades. No true quantitative measure has been developed for the pellicles of Bacilli, thus we cannot provide such measure. Importantly, the complexity and presence of *B. subtilis* pellicle formed at the air-medium interface directly correlate with plant colonization ability and biocontrol, as described in (11).

Minor issues:

1. The legend of figure 1A says: "plates were agitated at 50 rpm," while in the methods section, it is written that "plates were put on a laboratory shaker at 90 rpm".

A: Thanks for pointing out this mistake. The shaking condition of the EE setup was at 90 rpm. It is now corrected in the legend of Fig. 1A.

2. There is no explanation of how the authors differentiate single cells from chains (figure 2E and S7C). (membrane staining?)

A: Thanks for the comment. The differentiation method was applied according to previous research (10) without membrane staining. As described in the Methods part, cells were subjected to Cryo-SEM analysis, and then pools of Cryo-SEM images were analyzed with bacterial cells ($n > 50$). Cells in chains of two or more individuals were enumerated and categorized as 'chains', whereas one individual was categorized as single cell. The Cryo-SEM images allows clear determination of cell membranes, thus single cells are easy to follow.

3. The title of figure 6 is not clear "**An evolved isolate from lineage E and the *rhoGlu54stop* strain share similar gene expression patterns with the ancestor.**" The title suggests that Lineage E and *rhoGlu54stop* share the expression pattern **with** the ancestor. The truth is that they both share the expression pattern **differences compared** to the ancestor.

A: Thanks for pointing out this mistake. We have corrected it in the text as suggested.

References

1. Lereclus D, Arantès O, Chaufaux J, Lecadet MM. 1989. Transformation and expression of a cloned δ -endotoxin gene in bacillus thuringiensis. FEMS Microbiol Lett 60:211–217.
2. Sheppard AE, Poehlein A, Rosenstiel P, Liesegang H, Schulenburg H. 2013. Complete Genome Sequence of *Bacillus thuringiensis* Strain 407 Cry-. Genome Announc 1:158–170.
3. Perchat S, Talagas A, Poncet S, Lazar N, Li de la Sierra-Gallay I, Gohar M, Lereclus D, Nessler S. 2016. How Quorum Sensing Connects Sporulation to Necrotrophism in *Bacillus thuringiensis*. PLOS Pathog 12:e1005779.
4. Cardoso P de F, Perchat S, Vilas-Boas LA, Lereclus D, Vilas-Bôas GT. 2019. Diversity of the Rap–Phr quorum-sensing systems in the *Bacillus cereus* group. Curr Genet 65:1367–1381.

5. Buisson C, Gohar M, Huillet E, Nielsen-LeRoux C. 2019. *Bacillus thuringiensis* Spores and Vegetative Bacteria: Infection Capacity and Role of the Virulence Regulon PlcR Following Intrahaemocoel Injection of *Galleria mellonella*. *Insects* 10:129.
6. Brehélin M, Ramisse F, Gilois N, Hernandez E, Bourguet D, Salamiou S, Gominet M, Lereclus D, Ramisse F, Brehelin M, Bourguet D, Gilois N, Gominet M, Hernandez E, Lereclus D. 2000. The plcR regulon is involved in the opportunistic properties of *Bacillus thuringiensis* and *Bacillus cereus* in mice and insects. *Microbiology* 146:2825–2832.
7. Fagerlund A, Lindbäck T, Granum PE. 2010. *Bacillus cereus* cytotoxins Hbl, Nhe and CytK are secreted via the Sec translocation pathway. *BMC Microbiol* 10:304.
8. Candela T, Fagerlund A, Buisson C, Gilois N, Kolstø A, Økstad OA, Aymerich S, Nielsen-Leroux C, Lereclus D, Gohar M. 2019. CalY is a major virulence factor and a biofilm matrix protein. *Mol Microbiol* 111:mmi.14184.
9. Lin Y, Maróti G, Strube ML, Kovács ÁT. 2021. Adaptation and phenotypic diversification of *Bacillus thuringiensis* 407 biofilm are accompanied by a fuzzy spreader morphotype. *bioRxiv* 2021.09.03.458824.
10. Kearns DB, Losick R. 2005. Cell population heterogeneity during growth of *Bacillus subtilis*. *Genes Dev* 19:3083–3094.
11. Chen Y, Yan F, Chai Y, Liu H, Kolter R, Losick R, Guo JH 2013. Biocontrol of tomato wilt disease by *Bacillus subtilis* isolates from natural environments depends on conserved genes mediating biofilm formation. *Environ Microbiol* 15:848-864.

September 26, 2021

Prof. Ákos T Kovács
Technical University of Denmark
Department of Biotechnology and Biomedicine
Søltofts Plads 221
Kgs Lyngby 2800
Denmark

Re: mSystems00864-21R1 (Adaptation of *Bacillus thuringiensis* to plant colonization affects differentiation and toxicity)

Dear Prof. Ákos T Kovács:

Thank you for submitting your revised manuscript to mSystems. We have completed our review and I am pleased to inform you that, in principle, we expect to accept it for publication in mSystems. The reviewers were overall very satisfied with the revisions in your resubmitted manuscript. The only final request is that you explicitly refer to the BioRxiv manuscript you refer to in your rebuttal (9. Lin Y et al. 2021. Adaptation and phenotypic diversification of *Bacillus thuringiensis* biofilm... bioRxiv 2021.09.03.458824) and its relevance to your experiments within the main text of your revised manuscript. Addressing this comment is essential as this pre-print directly addresses the remaining major concern of this reviewer.

Preparing Revision Guidelines

Sincerely,

Elizabeth Shank

Editor, mSystems

Journals Department
Reviewer comments:

Reviewer #1 (Comments for the Author):

The authors have adequately dealt with the diverse suggestions that were made in the first round of review (by me and the other reviewer(s)) and I have no additional comments.

Reviewer #2 (Comments for the Author):

The Authors should refer to this comment in the corrected manuscript:

1. However, the experimental setting was devised for evolving increased biofilm formation and dispersal on polystyrene beads. It is not clear that the plant plays any role in this adaptation as "a plant," not just as a substrate for biofilm attachment. In addition, the condition of the experiment including shaking, and switching from plant to plant in each cycle, seem very artificial regarding root colonization in more natural settings. The authors need to strengthen the notion that the bacteria adapted to plant colonization, as this is the paper's main point. This may be done by monitoring root colonization of the evolved population in other conditions like agar plates and soil.

A: Thank you for pointing this issue. The EE setup in this study was indeed a modified version of the well know bead model that was previously used to test adaptation on abiotic surfaces. In the meantime, we have also experimentally evolved the bacteria in the bead system to test adaptation to abiotic biofilm formation. The work has been now deposited to the bioRxiv (9).

We reasoned that the microbe has distinct evolutionary patterns in these two EE setups. For example, in the bead system, evolved variants with the highest fitness mutated in a guanylyltransferase gene related to the cell surface properties. The mutation has very specific effects on the adhesiveness of the cells to the abiotic surfaces. On the contrary, the mutation in plant-associated evolved bacteria was more related to the carbohydrate utilization, the effects of which were more pervasive.

Furthermore, the mutational profiles of the microbes in these two setups were distinct as well. In plant-associated evolved variants, 70 SNPs were identified. For bead-associated variants, the number was 27. Importantly, there were few overlapping mutations among them in both setups.

Taking together, we believed this evolutionary pattern was specifically shaped by plant-associated biofilms. Thus, comparison of the two setups, biotic (plant) and abiotic (bead) and the derived conclusions in the two manuscripts clearly demonstrate distinct adaptation patterns.

Answer to Review comment

” The Authors should refer to this comment in the corrected manuscript:”

A: We have no address this remaining point as follows.

However, evolution on the root surface clearly differs from the adaptation pathway observed when *B. thuringiensis* is repetitively colonizing the abiotic surfaces of nylon beads (71). Evolved derivatives from the bead-adapted populations that displayed increased fitness compared to the ancestor, carried an insertion sequence in the guanylyltransferase gene, *rfbM*. RfbM seems to influence cell surface properties of *B. thuringiensis* and its mutation affects the cell adhesiveness to abiotic surfaces (71). On the contrary, mutations and transcriptional changes of plant-associated evolved derivatives were related to carbohydrate utilization, the effects of which were more pervasive. Furthermore, the mutational profiles of the evolved isolates in these two setups were distinct as well (71).

September 27, 2021

Prof. Ákos T Kovács
Technical University of Denmark
Department of Biotechnology and Biomedicine
Søltofts Plads 221
Kgs Lyngby 2800
Denmark

Re: mSystems00864-21R2 (Adaptation of *Bacillus thuringiensis* to plant colonization affects differentiation and toxicity)

Dear Prof. Ákos T Kovács:

Your manuscript has been accepted, and I am forwarding it to the ASM Journals Department for publication. For your reference, ASM Journals' address is given below. Before it can be scheduled for publication, your manuscript will be checked by the mSystems senior production editor, Ellie Ghatineh, to make sure that all elements meet the technical requirements for publication. She will contact you if anything needs to be revised before copyediting and production can begin. Otherwise, you will be notified when your proofs are ready to be viewed.

As an open-access publication, mSystems receives no financial support from paid subscriptions and depends on authors' prompt payment of publication fees as soon as their articles are accepted. =

Publication Fees:

We recognize that the video files can become quite large, and so to avoid quality loss ASM suggests sending the video file via <https://www.wetransfer.com/>. When you have a final version of the video and the still ready to share, please send it to Ellie Ghatineh at eghatineh@asmusa.org.

Sincerely,

Elizabeth Shank
Editor, mSystems

Journals Department
Fig S7: Accept
Fig S3: Accept
Fig S5: Accept
Fig S6: Accept
Fig S1: Accept
Fig S2: Accept
Dataset S2: Accept
Dataset S1: Accept
Fig S4: Accept
Dataset S3: Accept